



# Resolution Dependence and Biases in Cold and Warm Frontal Extreme Precipitation over Europe in CMIP6 and EURO-CORDEX Models

Armin Schaffer[1], Tobias Lichtenegger[1], Albert Ossó[1], and Douglas Maraun[1]

[1]Wegener Center for Climate and Global Change, University of Graz, Graz, Austria

**Correspondence:** Armin Schaffer (armin.schaffer@uni-graz.at)

**Abstract.** Atmospheric cold and warm fronts are a major driver of extreme precipitation over Europe. To assess future changes in extreme weather, it is therefore essential to understand how frontal systems respond to a warming climate. This requires the analysis of climate model projections. A crucial first step is a process-based evaluation of frontal dynamics in present-day simulations, as this increases confidence in the models and the reliability of their future projections.

In this study, we compare the representation of frontal frequencies, frontal extreme precipitation, and frontal structure in the CMIP6 and EURO-CORDEX ensembles, using ERA5 as a reference. To assess the added value of higher resolution, we analyze the models on their native grids and compare them with ERA5 data remapped to similar resolutions.

We found that all models exhibit substantial biases in frontal frequencies and associated extreme precipitation, which are possibly related to storm-track position biases and an underrepresentation of land–atmosphere interactions. Warm frontal extremes are generally better captured than cold frontal extremes. Increasing model resolution leads to significant improvements for cold frontal biases, whereas warm frontal biases remain largely unaffected. The analysis of frontal structures supports this interpretation: while synoptic-scale conditions are well represented across models, mesoscale gradients and circulation patterns exhibit a pronounced sensitivity to grid spacing. Because warm fronts extend over larger spatial scales, they are already reasonably well simulated at coarse resolution. Cold fronts, by contrast, are governed by smaller-scale processes and therefore show notable improvements at higher resolution.

These findings provide an important step toward evaluating climate models in their ability to simulate extreme weather phenomena. While warm frontal extremes appear robust across model resolutions, reliable simulations of cold frontal extremes require higher-resolution models to adequately capture their dynamics and associated extreme precipitation.

## 1 Introduction

Extreme weather in the mid-latitudes is frequently caused by atmospheric fronts (Catto et al., 2012). Cold fronts are typically associated with strong wind gusts and intense short-duration precipitation, often leading to localized flooding. In contrast, warm frontal precipitation is generally more widespread but less intense, potentially causing flooding across larger catchments.

Numerous studies have demonstrated the connection between frontal systems and extremes in Europe. For example, half of all hail events in Switzerland have been linked to cold fronts (Schemm et al., 2016), while in Germany convective cells are



twice as likely to form in the vicinity of cold fronts during the warm season (Pacey et al., 2023). The proportion of precipitation associated with all fronts over Europe range from 30–80 %, with strong regional and seasonal differences (Catto et al., 2012; Hénin et al., 2019; Rüdisühli et al., 2020). For extreme precipitation events, the estimated contribution increases to 60–90 % (Catto and Pfahl, 2013).

Given their central role in generating high-impact weather, future changes in frontal systems need to be studied in detail
to increase confidence in projections of extremes (Collins et al., 2018; Marotzke et al., 2017). Therefore, it is essential to first assess how well climate models represent atmospheric fronts in the current climate and, based on these process-based evaluations, assess projections of extremes (IPCC, 2023).

In our previous work, we examine the drivers of cold frontal extremes (Schaffer et al., 2024) and analyze the cold frontal life cycle in reanalysis data (Lichtenegger et al., 2025). Building on these findings and methodologies, we now assess the
performance of a range of climate models. Catto et al. (2014) were the first to evaluate front frequencies in climate models under both present and future conditions, finding good agreement between ERA-Interim and CMIP5 datasets. They also projected decreases in both the frequency and intensity of fronts across many regions under the RCP8.5 scenario.

Here, we evaluate the Coupled Model Intercomparison Project Phase 6 (CMIP6) dataset (Eyring et al., 2016), together with the latest generation of the European Centre for Medium-Range Weather Forecasts atmospheric reanalysis (ERA5) (Hersbach
et al., 2020) as a reference. Our analysis extends previous work by investigating the added value of higher-resolution models in simulating atmospheric fronts and by quantifying biases in the associated extreme precipitation. To explore a broader range of model resolutions, we further analyze the Coordinated Downscaling Experiment over Europe (EURO-CORDEX) dataset (Jacob et al., 2014). Extending the analysis from GCMs to RCMs enables us to evaluate the benefits of resolving fronts at the 10 km-scale. A key focus is to link potential biases in extreme precipitation to deficiencies in the representation of the frontal
structure and processes. To this end, for the first time, we perform an analysis of frontal cross-sections in a climatological model study.



**Table 1.** Overview of all datasets used in this study, including horizontal resolution, number of available pressure levels and associated sub-ensemble classification.

| RCM | GCM | Resolution [°] | Number of levels | Sub-ensemble |
|---|---|---|---|---|
| | ERA5 | 0.25 × 0.25 | 10[1] | |
| | ERA5 0.9° | 0.9 × 0.9 | 10[1] | |
| | ERA5 1.25° | 1.25 × 1.25 | 10[1] | |
| | ERA5 2° | 2 × 2 | 10[1] | |
| ALADIN63 | CNRM-CERFACS-CNRM-CM5 | 0.11 × 0.11 | 5[2] | CORDEX |
| ALADIN63 | MOHC-HadGEM2-ES | 0.11 × 0.11 | 5[2] | CORDEX |
| ALADIN63 | MPI-M-MPI-ESM-LR | 0.11 × 0.11 | 5[2] | CORDEX |
| ALADIN63 | NCC-NorESM1-M | 0.11 × 0.11 | 5[2] | CORDEX |
| COSMO-crCLIM-v1-1 | CNRM-CERFACS-CNRM-CM5 | 0.11 × 0.11 | 8[3] | CORDEX |
| COSMO-crCLIM-v1-1 | MOHC-HadGEM2-ES | 0.11 × 0.11 | 8[3] | CORDEX |
| COSMO-crCLIM-v1-1 | MPI-M-MPI-ESM-LR | 0.11 × 0.11 | 8[3] | CORDEX |
| COSMO-crCLIM-v1-1 | NCC-NorESM1-M | 0.11 × 0.11 | 8[3] | CORDEX |
| COSMO-crCLIM-v1-1 | ICHEC-EC-EARTH | 0.11 × 0.11 | 8[3] | CORDEX |
| RCA4 | CNRM-CERFACS-CNRM-CM5 | 0.11 × 0.11 | 8[3] | CORDEX |
| RCA4 | MOHC-HadGEM2-ES | 0.11 × 0.11 | 8[3] | CORDEX |
| RCA4 | MPI-M-MPI-ESM-LR | 0.11 × 0.11 | 8[3] | CORDEX |
| RCA4 | NCC-NorESM1-M | 0.11 × 0.11 | 8[3] | CORDEX |
| RCA4 | ICHEC-EC-EARTH | 0.11 × 0.11 | 8[3] | CORDEX |
| RCA4 | IPSL-IPSL-CM5A-MR | 0.11 × 0.11 | 8[3] | CORDEX |
| | GISS-E2-1-G | 2 × 2.5 | 33 / 23[4] | CMIP6 180 km |
| | NorESM2-LM | 1.875 × 2.5 | 33 / 23[4] | CMIP6 180 km |
| | MPI-ESM1-2-LR | 1.875 × 1.875 | 33 / 23[4] | CMIP6 180 km |
| | AWI-ESM-1-1-LR | 1.8653 × 1.875 | 33 / 23[4] | CMIP6 180 km |
| | MPI-ESM-1-2-HAM | 1.8653 × 1.875 | 33 / 23[4] | CMIP6 180 km |
| | IPSL-CM6A-LR-INCA | 1.25 × 2.5 | 33 / 23[4] | CMIP6 120 km[5] |
| | IPSL-CM6A-LR | 1.25 × 2.5 | 33 / 23[4] | CMIP6 120 km[5] |
| | MIROC6 | 1.4 × 1.4 | 33 / 23[4] | CMIP6 120 km |
| | MRI-ESM2-0 | 1.125 × 1.125 | 33 / 23[4] | CMIP6 120 km |
| | TaiESM1 | 0.9424 × 1.25 | 33 / 23[4] | CMIP6 90 km |
| | NorESM2-MM | 0.9357 × 1.25 | 33 / 23[4] | CMIP6 90 km |
| | CMCC-CM2-SR5 | 0.9357 × 1.25 | 33 / 23[4] | CMIP6 90 km |
| | CMCC-ESM2 | 0.9357 × 1.25 | 33 / 23[4] | CMIP6 90 km |
| | MPI-ESM1-2-HR | 0.9375 × 0.9375 | 33 / 23[4] | CMIP6 90 km |
| | EC-Earth3 | 0.7031 × 0.7031 | 33 / 23[4] | CMIP6 90 km |

[1] 1000, 925, 900, 850, 700, 600, 500, 400, 300, 200 hPa

[2] 925, 850, 700, 500, 200 hPa

[3] 925, 850, 700, 600, 500, 400, 300, 200 hPa

[4] 1000 – 200 (25 hPa steps), with geopotential missing 725 – 275 (50 hPa steps).

[5] The IPSL model is included in the CMIP6 120 km sub-ensemble because its results aligned more closely with that group than with others.



## 2 Data

We analyze 15 GCMs from the CMIP6 ensemble (Eyring et al., 2016), 15 RCM simulations from EURO-CORDEX (Jacob et al., 2014), and the ERA5 reanalysis dataset (Hersbach et al., 2020) for the period 1970 – 2005. Within the EURO-CORDEX
ensemble, three different RCMs are each driven by between four and six GCMs. All simulations providing the necessary 6-hourly, three-dimensional fields of temperature, humidity, wind, geopotential height, and precipitation from the ensembles have been selected. A complete list of all models used is provided in Table 1.





**Figure 1.** Front frequency in ERA5 at (a, b) 0.25°, (c, d) 2°, (e, f) 1.25°, and (g, h) 0.9° resolution for (a, c, e, g) cold and (b, d, f, h) warm fronts. Panels (a, b) show the number of timesteps with front occurrence per year while (c–h) depict the relative difference compared to native-resolution ERA5.



## 2.1 Resolution effect

To evaluate the added value of higher spatial resolution, all analysis is performed on the native model grids. However, the analysis itself is inherently impacted by resolution because coarser grids introduce smoothing to all physical fields. The averaging not only reduces the representation of smaller-scale physics, but for precipitation fields it potentially hides its extremeness. To improve consistency, all data are spectrally filtered prior to the analysis. However, the horizontal grid resolution continues to strongly impact the results. To illustrate this effect, a comparison of front frequencies in ERA5 compared with ERA5 remapped to three coarser resolutions (2°, 1.25° and 0.9°) is shown in Fig. 1. The error introduced by resolution is of the same order as the model biases. Previous studies avoided potential methodological biases from resolution disparities by remapping all data to the same coarse grid (e.g. Catto et al. (2014); King et al. (2024)).

In this study, we minimize methodological biases related to resolution differences, while assessing the added value of higher resolutions, by using a similar approach as described in Volosciuk et al. (2015): ERA5 is remapped to the three aforementioned coarser resolutions of 2° × 2°, 1.25° × 1.25°, and 0.9° × 0.9°. Based on their resolution, the CMIP6 models are classified into three sub-ensembles: coarse (180 km), mid (120 km), and high resolution (90 km). These sub-ensembles are compared with the remapped ERA5 data with the closest mean horizontal grid spacing. This step minimizes errors arising from resolution differences when comparing datasets. The EURO-CORDEX models are analyzed similarly on the native grid and evaluated against the native ERA5 data. Because ERA5 has a coarser grid than EURO-CORDEX, some methodological bias is introduced to this analysis. For comparison and plotting purposes, all results are remapped to a uniform 0.25° × 0.25° resolution.

The mid-resolution CMIP6 sub-ensemble consists of four models, of which two are from the IPSL. Unlike the other models, these have distinct latitudinal and longitudinal grid sizes. As a consequence, their results align best with those of mid-resolution CMIP6 models. For that reason, they are classified with this sub-ensemble. Because half of the sub-ensemble is made up of IPSL models, they have a strong impact on the results. These effects are mentioned in the result section and are discussed in more detail in the Appendix, but the evaluation of specific causes is beyond the scope of this study.

## 3 Methods

The methods applied in this study closely follow those described in Schaffer et al. (2024). An overview of the approach, including all modifications, is provided here. For a more detailed explanation, we refer the reader to the previous study.

### 3.1 Front detection

Fronts are detected by applying a threshold to the smoothed equivalent potential temperature gradient ($\nabla\theta_e$) field at 850 hPa. The threshold is defined as the seasonal spatial mean plus one standard deviation of $\nabla\theta_e$ over the North Atlantic region ($20°W - 12°W, 40°N - 58°N$) for the period $1970 - 2005$, computed separately for each model. This region was selected because it is covered by all datasets, lies within the climatological storm track, and is free from orographic interference.





The final frontal points are identified as local maxima in $\nabla \theta_e$ where the Thermal Front Parameter (TFP) is closest to zero. Based on the cross-frontal wind speed ($u_f$), these points are classified into cold fronts ($u_f > 1.5 \text{ m s}^{-1}$) and warm fronts ($u_f < -1.5 \text{ m s}^{-1}$). Points with weaker $u_f$ are excluded to ensure that only mobile synoptic-scale fronts are retained. Connected frontal points are grouped into contiguous frontal objects. To exclude non-synoptic features, a minimum length threshold of 500 km is applied.

The detected fronts still contain some unwanted objects. Along coasts and mountain ranges (e.g., the Norwegian coast, the western Balkan coast), humidity differences can lead to the detection of false cold fronts, which are not relevant for our analysis. We remove these objects by comparing the angles of the $\nabla \theta_e$ and geopotential height gradient vectors. In synoptic-scale cold fronts, these vectors typically point in the same direction. By applying a threshold on the absolute angle difference of greater than 120°, we not only remove non-frontal objects but also exclude back-bent occlusion fronts from the cold front analysis. Similarly, strong humidity gradients at the warm-side boundary of the warm conveyor belt are often falsely detected as warm fronts. To filter these objects, we require a positive potential temperature difference between points located 300 km ahead and 300 km behind of the warm frontal points at 850 hPa. Because the warm conveyor belt generally exhibits higher equivalent potential temperatures but lower potential temperatures than the air farther ahead of the cold front, the criterion effectively removes many of these unwanted detections.

### 3.2 Frontal frequency and precipitation

After the detection, the frontal data consisting of grid points labeled either 1 or -1 depending on the front type. In this state, it is not possible to compare the frequencies of the different models due to the dependence of the frontal width on the resolution of the grid. Using on a grid-factor (e.g. number of high-resolution grid points per low-resolution grid point) is also not possible, because of the varying extent and curvature of fronts in higher resolution data. To make the frequencies comparable, we define a frontal area as a circular region with a 300 km radius centered on each frontal point. The resulting objects have consistent width throughout all data. These areas are masked and subsequently summed up to estimate the frequency of fronts.

Precipitation is considered frontal if it occurs within the defined frontal area. We further classify precipitation as extreme if the 6-hourly total exceeds the 99.5th percentile at each grid point, corresponding to a return period of approximately 50 days. Following Hénin et al. (2019), precipitation is attributed to cold or warm fronts when it falls within the radius of both front types. In such cases, precipitation is partitioned based on the relative number of grid points associated with each front type: the total precipitation is multiplied by the fraction of each type's frontal points relative to the combined total. By summing the total and frontal extreme precipitation, we compute the fraction of extreme precipitation associated with each front type.

### 3.3 Frontal composites

The precipitation associated with each frontal object is evaluated by averaging values within the 200 km front segment exhibiting the highest standardized precipitation. These segments do not need to be continuous but are composed of frontal points that, based on the model grid, together approximate a total length of 200 km. To avoid multiple sampling of the same event, only one front per 24-hour period is included in the analysis. The top 10 % of frontal objects, ranked by standardized precipi-





tation, are then used to generate the composites. The regions used to evaluate these extreme fronts follow the same definitions as in Schaffer et al. (2024). To ensure a more focused and comprehensive analysis, we combine the extreme fronts from all three regions and seasons into a single dataset, from which composites are subsequently calculated. The frontal cross-section composites are generated by extracting 1200 × 1200 km atmospheric fields centered on the frontal objects. The position of

120 each frontal object is determined by the frontal point with the median standardized precipitation within the selected 200 km segment. The extracted fields are rotated into the cross-frontal direction before calculating the composites.

## 4 Results

### 4.1 Frontal frequency

Before evaluating model biases in front frequency and the associated extremes, we want to give a detailed look at the effect

of horizontal resolution on the detection for ERA5. To this end we compare ERA5 on the native grid to the three coarsened remaps, which will act as a reference to the model data. Fig. 1a–b show the absolute front frequency based on native-resolution ERA5, which is qualitatively consistent with the hourly frequencies reported in Schaffer et al. (2024). Quantitative differences arise from the differing temporal resolutions, as well as from the adapted frequency evaluation methodology. In Fig. 1c–h, the previously discussed strong relationship between grid resolution and front frequency is evident. The coarsest resolution data

(2°) show over 50 % fewer cold and warm fronts across large parts of the study domain. The bias is substantially reduced at higher resolutions (1.25° and 0.9°). In some regions, higher frequencies in the remap data compared to the native data can be observed (e.g., along the coast of Greenland, the Balkans, Anatolia). Positive resolution biases in these areas are likely due to the influence of orography, which becomes increasingly smoothed at lower resolutions, thereby enabling more continuous front detection in mountainous regions.





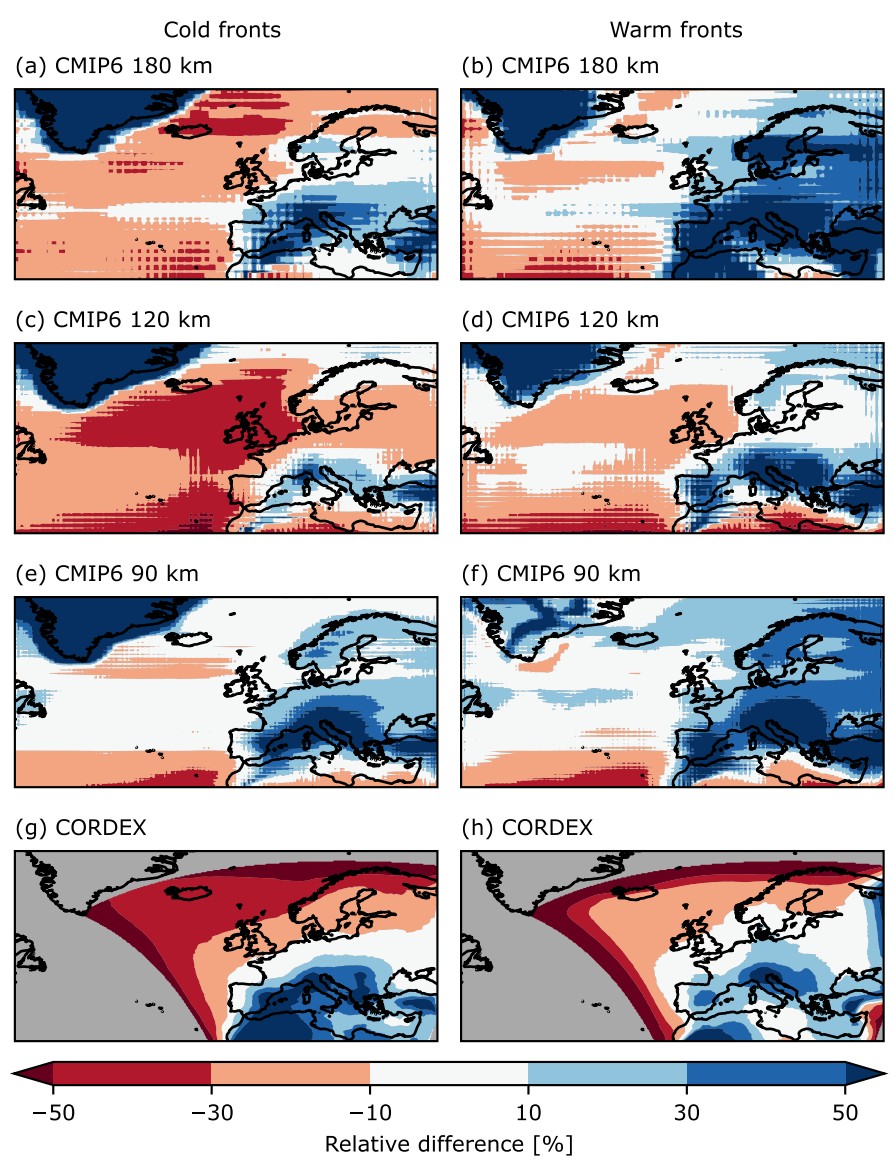

**Figure 2.** Relative front frequency bias of the (a, b) coarse CMIP6, (c, d) mid CMIP6, (e, f) fine CMIP6 and (g, h) CORDEX models compared to the respective ERA5 remap for (a, c, e, g) cold and (b, d, f, h) warm fronts. Grey indicate areas outside the EURO-CORDEX domain. The pronounced negative biases of CORDEX near the northern and western domain boundaries are caused by the 500 km minimum front length threshold, which excludes valid fronts that extend beyond the domain boundaries.



Model biases in frontal precipitation may be explained primarily by underlying frontal frequency biases. Thus, in the following the relative front frequency biases of the different model ensembles are shown in Fig. 2. In general, the models simulate lower front frequencies over the North Atlantic and higher frequencies over continental Europe. Notably, positive biases are found over orographically complex regions such as the Alps and Anatolia, as well as on the leeward sides of major mountain ranges (e.g., the Iberian Peninsula and Scandinavia). These patterns are consistent across all model resolutions and do not

improve when mountains are represented in more detail, suggesting that the smoothed orography is not the main cause. A reduction of boundary-layer friction in the models could intensify frontal circulation and enhance frontogenesis, which would strengthen $\nabla\theta_e$ and could increase the likelihood of detecting fronts.

Looking more into the individual sub-ensemble results, it can be seen that CORDEX exhibits a clear positive bias over southern Europe and a negative bias over northern Europe for cold and warm fronts alike (Fig. 2g–h). Previous studies have

145 documented an overly zonal storm track in CMIP5 simulations (Harvey et al., 2020; Priestley et al., 2020), which may contribute to these regional differences in the EURO-CORDEX ensemble. A similar storm track bias may also influence CMIP6 simulations, although the same studies have indicated that the latest CMIP version has improved their performance in this regard. The elevated front frequencies in a zonal band over the North Atlantic (approximately $43-47°$N) in CMIP6 models may still be the result of the residual error in the storm track position.

The effect of these biases can be seen in both cold and warm frontal frequencies. Comparing cold and warm fronts in more detail reveals that warm fronts are generally detected more frequently in the models. Over the North Atlantic, they show smaller differences compared to ERA5, whereas over continental Europe, the differences are larger. The ratio of cold to warm fronts remains relatively constant across model resolutions.





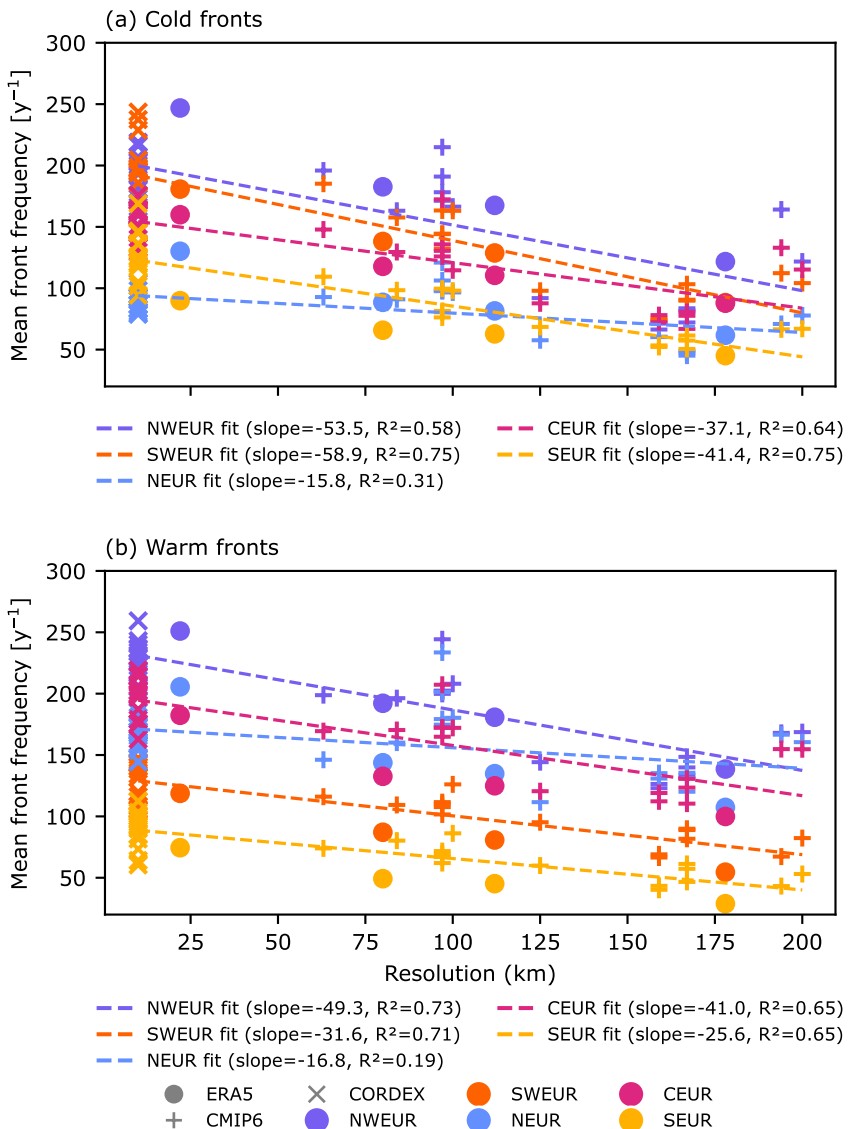

**Figure 3.** Regression of model resolution with mean front frequency over selected regions for (a) cold and (b) warm fronts. The colored (•) indicate ERA5 and its remaps, (x) CMIP6 and (+) CORDEX models. The colored dashed lines show the regression lines, with the values of the slopes and regression coefficients ($R^2$) in the legends. The colors indicate the regions. The y-axis shows the mean front frequency values. The resolution values depicted on the x-axis are equal to the the geometric mean horizontal grid lengths at 50°N.



Following the bias analysis, we investigate the effect of mean horizontal resolution on front frequencies across specific
regions, as defined in Fig. A1. The regression between model resolution and front frequency is shown in Fig. 3 for cold (a) and
warm (b) fronts. Across all datasets, a strong negative relationship is found: smaller grid sizes are associated with higher front
frequencies. Increasing the resolution by 100 km results in the detection of approximately 15–60 additional fronts per year
within each region, corresponding to an increase of $8-46$ %. Cold fronts show slightly steeper regression slopes, ranging from
16–60 fronts per year per 100 km ($12-46$ %), compared to warm fronts, which range from 17–50 fronts per year per 100 km
($8-34$ %). Overall, these results suggest that cold frontal frequencies exhibit a stronger sensitivity to horizontal resolution
compared to warm fronts.

The north-south bias in the CORDEX models, previously related to the storm track position, is also evident in the regression
results. The NEUR and NWEUR region have lower mean front frequency values than expected from the regression. For the
NEUR region the strong domain boundary error may also have an impact. When CORDEX is excluded, the relationship
becomes stronger, with the steepness of the slope increasing from 54 and 16 to 86 and 40 additional fronts per 100 km,
respectively (Fig. A2). Furthermore, the slopes for SEUR decrease from 41 and 26 to 29 and 22 fronts per 100 km, indicating
that CORDEX tends to underestimate frontal frequencies in northern regions while overestimating them in southern Europe.

These results confirm and extend the resolution-related findings from Fig. 1 to the full ensemble of climate model data. The
resolution dependence of front detection is a key factor in evaluating both the bias in frontal extreme precipitation and the
potential resolution effects on it.





**Figure 4.** Same as Fig. 1, but for frontal extreme precipitation.





## 4.2 Frontal precipitation

Similarly to the front frequency analysis, the effect of resolution on frontal extreme precipitation is the first important step to evaluate biases. Differences between the remapped ERA5 data and the native resolution data again reveal a strong dependence on grid size. The coarsest grid shows a reduction of 15–75 % in frontal extreme precipitation (Fig. 4c–d). Similar to the pattern observed in the front frequency analysis, the bias decreases with increasing resolution (Fig. 4e–f). The largest negative differences appear in the southern part of the study domain, which also exhibit strong biases in front frequency, while areas with positive differences in frontal precipitation tend to coincide with regions where more fronts are detected. A similar resolution dependence is found for the total amount of extreme precipitation (i.e., not only frontal), which decreases on coarser grids due to a dampening of the tail of the precipitation distribution (Fig. A3).

When comparing extreme precipitation associated with each front type, warm fronts exhibit smaller deviations from the native-resolution data than cold fronts. Part of this effect could be attributed to the fact that precipitation near the occlusion point tends to be classified more often as warm frontal in lower-resolution data. The likely reason is that smoother $\theta_e$ and wind fields in coarser grids shift the detected position of cold fronts farther from their intersection with warm fronts. Since high-intensity precipitation frequently occurs near the occlusion point, a larger fraction of precipitation may therefore be attributed to warm fronts in coarser-resolution data. However, the main cause is likely the broader area that warm frontal precipitation typically spans compared to the narrow cold frontal rain bands. It is therefore better represented at coarser resolutions. This finding highlights the importance of high-resolution GCMs for capturing the intensity of sub-daily cold frontal precipitation. Warm frontal precipitation, on the other hand, is shown to be less sensitive to grid size.



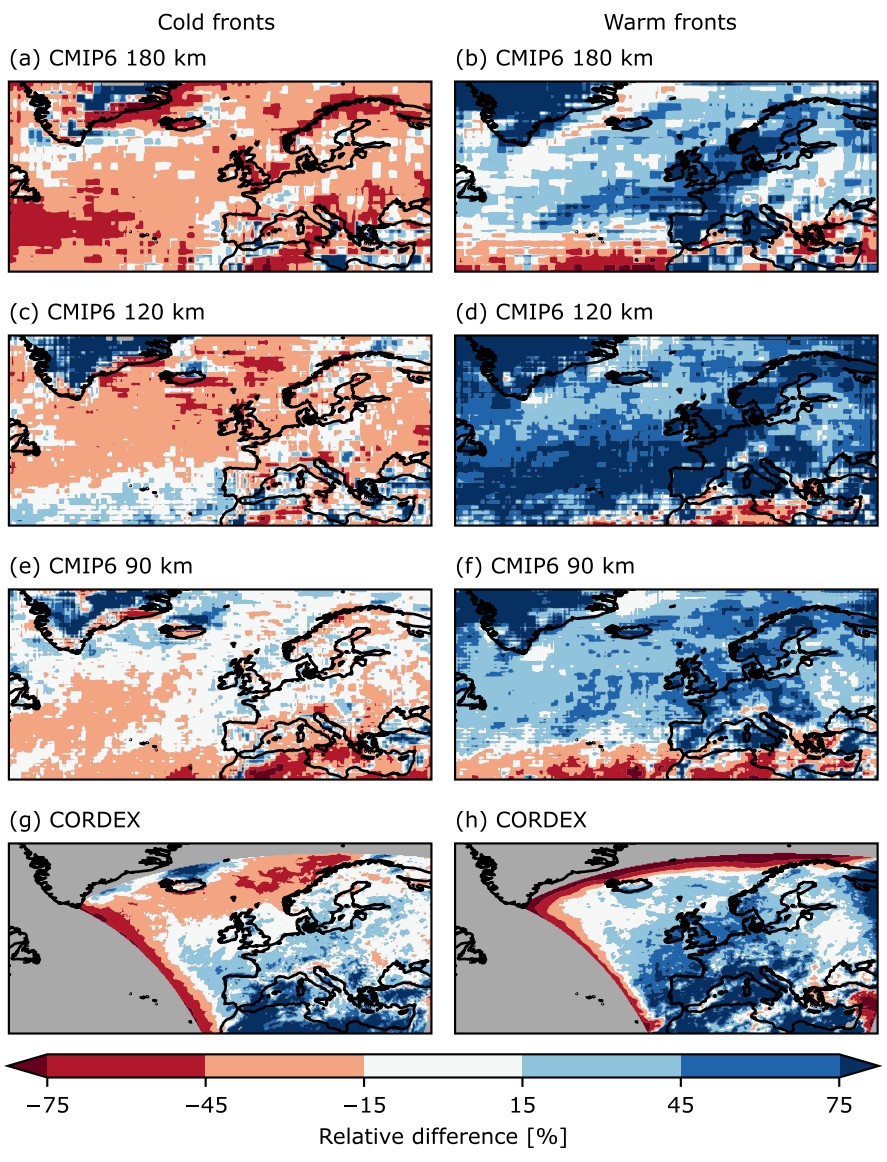

**Figure 5.** Same as Fig. 2, but for frontal extreme precipitation bias.



The model data similarly favor warm over cold frontal extreme precipitation. In Fig. 5, a general negative bias is evident
for cold frontal precipitation, while a strong positive bias is seen for warm frontal precipitation. When both front categories
are combined, the deviations from ERA5 are reduced to within $\pm 15$ % across most areas in CMIP6. The pronounced positive
bias over the Mediterranean and northern Africa in the CORDEX ensemble is mainly driven by the increase in front frequency,
likely linked to the aforementioned storm track bias. This interpretation is supported by the average precipitation per front bias,
which is comparable to the other model ensembles (Fig. A4).

Once again, cold fronts seem to benefit from higher resolution in the GCMs: with decreasing grid size, the cold frontal
precipitation bias is reduced across much of the study domain. The reduction suggests an improved representation of cold
frontal processes and associated rainbands in higher-resolution simulations. In contrast, the warm frontal bias remains largely
consistent across all sub-ensembles. Even over the Atlantic where warm front detection is substantially lower than in ERA5,
the models simulate higher extreme precipitation associated with warm fronts. The CMIP6 120 km sub-ensemble displays
very high warm frontal precipitation over both the ocean and the continent. A detailed analysis of the constituent models of the
sub-ensemble is presented in the Appendix (Fig. A5).

An analysis of the fraction of extreme precipitation associated with cold and warm fronts supports the conclusion, that mainly
cold frontal precipitation benefits from higher resolutions (Fig. A6). In CMIP6, absolute extreme precipitation values increase
with resolution, but the fraction only increases for cold fronts (i.e. the fraction difference to ERA5 decreases). The increase in
warm frontal precipitation is driven by an increase in total extreme precipitation, related to the reduced grid averaging effect,
not to an improved physical representation. However, better-resolved cold fronts benefit more strongly, highlighting the added
value of finer grids.




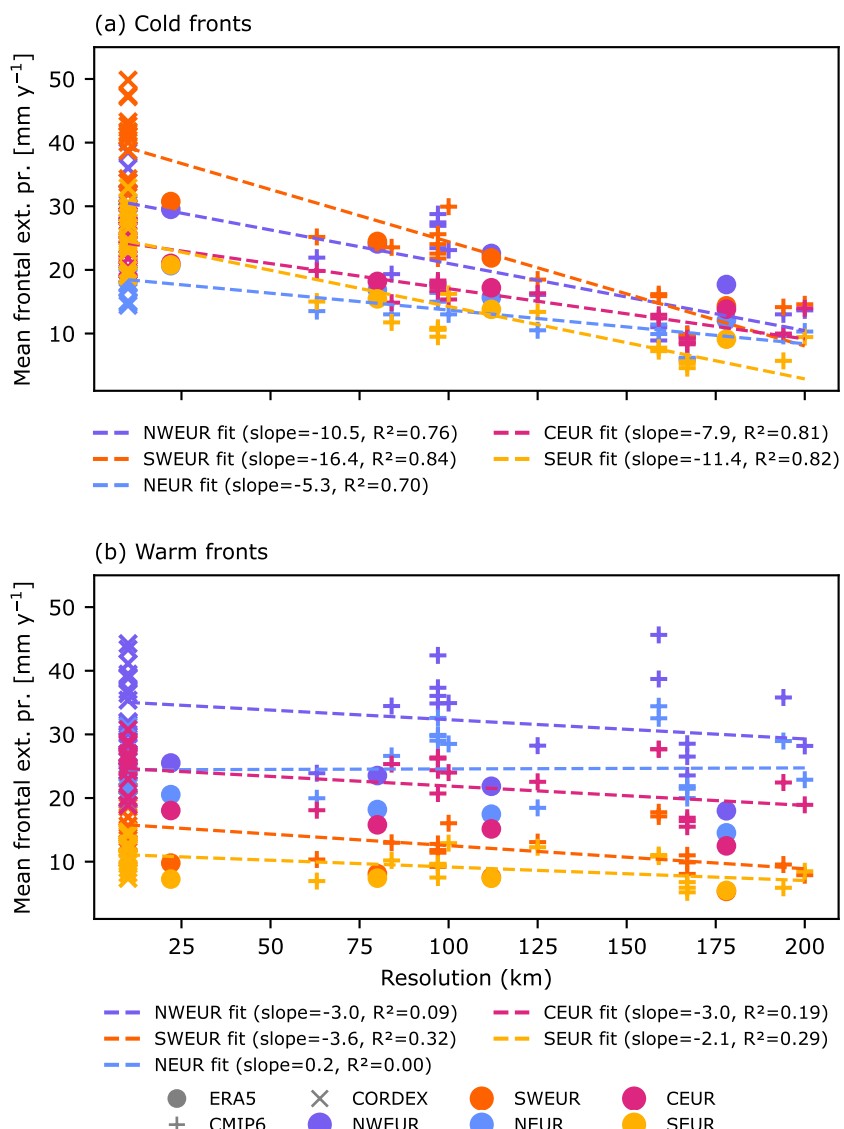

**Figure 6.** Same as Fig. 3, but showing the regression of model resolution with mean frontal extreme precipitation over selected regions.





To quantify this resolution effect more explicitly, we analyze the relationship between grid spacing and frontal extreme precipitation. Figure 6a illustrates the positive influence of resolution on cold frontal extreme precipitation. For every 100 km decrease in grid spacing, an average increase of $5-16$ mm of extreme precipitation per year is detected per region. While this relationship may partly result from higher cold front frequencies at finer resolutions, an improved representation of smaller-scale frontal processes is also likely to contribute. Restricting the analysis to CMIP6 and ERA5 data further enhances the consistency and alignment of regional patterns (Fig. A7a).

In contrast, warm frontal precipitation shows only a weak negative correlation with resolution (Fig. 6b). When CORDEX models are excluded, this relationship disappears entirely (Fig. A7b), even though higher-resolution models detect warm fronts more frequently, as shown in Fig. 3b. This finding further supports the idea that warm frontal precipitation and associated rainbands are already well captured by current-generation GCMs.



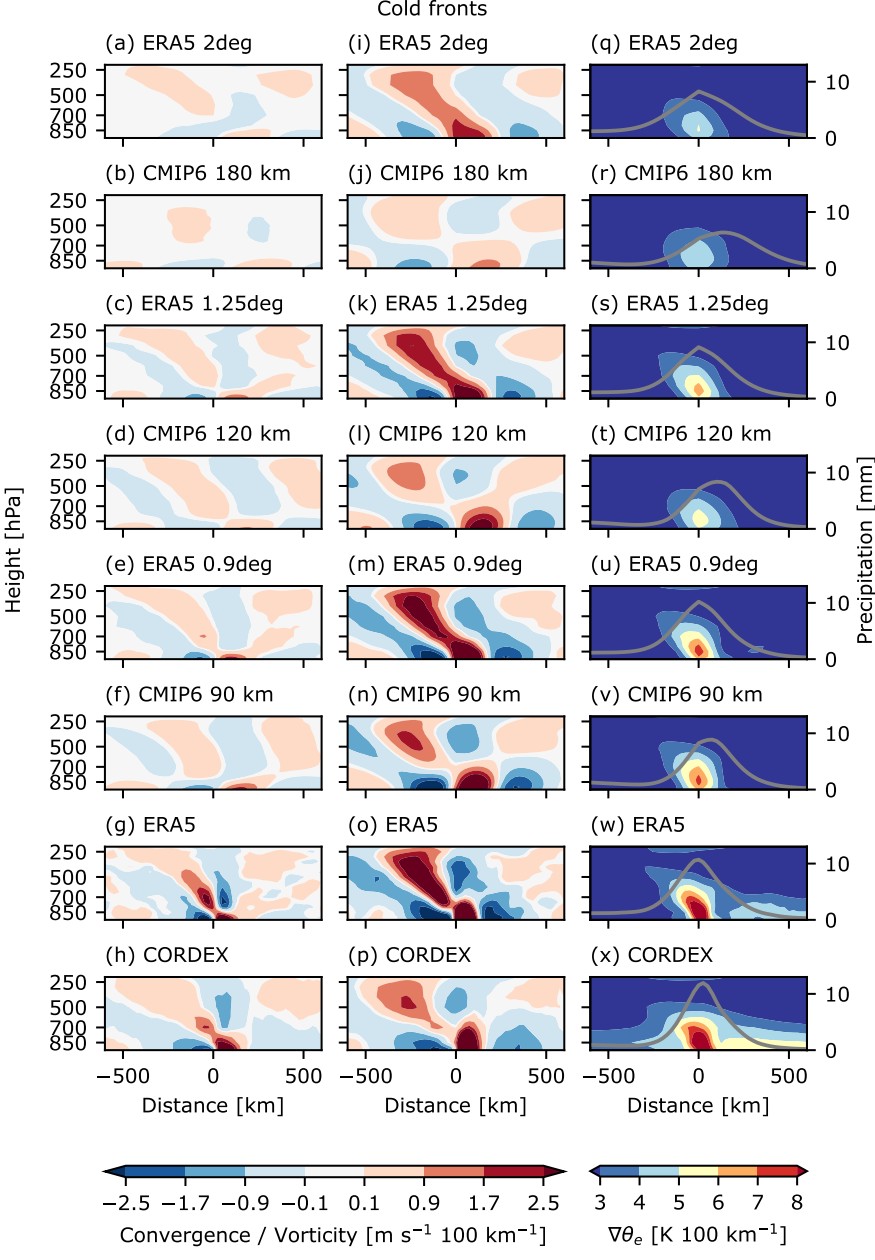

**Figure 7.** Cross-section composites of cold fronts. The x-axis depicts the cross-frontal distance from 600 km behind to 600 km ahead of the detected front at 850 hPa. The y-axis depicts the vertical pressure levels from 925–200 hPa and in (q–x) additionally the precipitation in mm. The filled contours display (a–h) mesoscale horizontal convergence in $\mathrm{m\,s^{-1}\,100\,km^{-1}}$, (i–p) mesoscale horizontal vorticity in $\mathrm{m\,s^{-1}\,100\,km^{-1}}$ and (q–x) $\nabla\theta_e$ in $\mathrm{K\,100\,km^{-1}}$. The contour in (q–x) illustrates the distribution of precipitation. Each panel row represents the composite of ERA5 or a sub-ensemble.



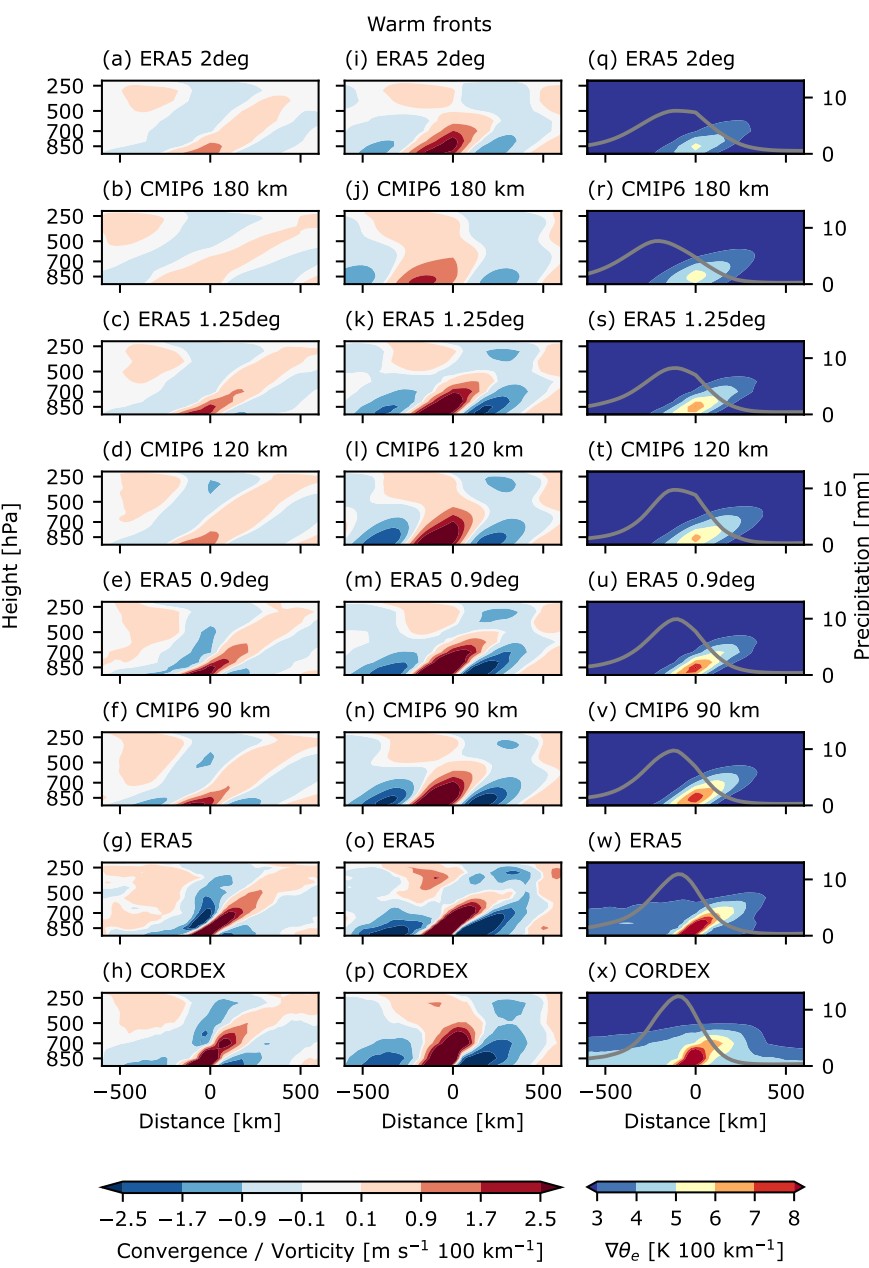

**Figure 8.** Same as 7, but for warm front composites.



### 4.3 Frontal composites

To understand why cold frontal extreme precipitation is particularly sensitive to model resolution, we examine the frontal
structure and associated circulation patterns. Our goal is to identify structural differences across models that may explain the
resolution–precipitation relationship observed in the previous section. For this purpose, we analyze cross-sections of composites for extreme cold (Fig. 7) and warm fronts (Fig. 8). Note that these composites are based on fronts selected independently
from those associated with the extreme precipitation events discussed in the previous section, but by following the method
described in Schaffer et al. (2024).

At the synoptic scale, the composite fields of temperature, humidity, and large-scale circulation show good agreement between ERA5 and all model ensembles. However, mesoscale gradients (Fig. 7q–x, Fig. 8q–x) and circulation features (Fig. 7a–p,
Fig. 8a–p) display a pronounced sensitivity to horizontal resolution. This is expected as finer grids resolve smaller-scale variability, which can enhance the representation of the frontal structure. Improved depiction of these fine-scale features, in turn,
increases precipitation variability and allows models to better capture extreme events.

Overall, the frontal structure is well captured across all models when differences in resolution are considered. Mesoscale
circulations, however, show some biases. The maximum vorticity is approximately 20 % lower in all models, and the maximum
convergence ranges from $-30$ % in coarse-resolution models to $+30$ % in CORDEX. Although these differences are evident,
they do not indicate fundamental structural errors. Nevertheless, limited representation of small-scale dynamics by coarse
models likely diminishes the intensity and organization of precipitation bands, a factor especially critical for cold frontal
extreme precipitation.

Comparing cold and warm fronts reveals systematic differences in precipitation characteristics (Fig. 7q–x, Fig. 8q–x). Warm
fronts exhibit larger total precipitation in the composites, though not necessarily higher peak intensities, consistent with the
warm front precipitation biases over Europe. Although total precipitation varies between models, there is no clear dependence
on resolution. In contrast, for both cold and warm fronts, the distribution of precipitation narrows with increasing resolution,
reflecting a higher ability of finer grids to resolve localized extremes.

Furthermore, the lower-level convergence and upper-level divergence patterns characteristic of strong updrafts illustrate the
smaller-scale nature of cold fronts. While warm fronts have an extended region of convergence, tilting forward with $\nabla \theta_e$, cold
fronts exhibit a comparatively narrow updraft region. The larger horizontal extent of warm fronts, characterized by a stronger
tilt, may be the reason for improved representation in coarse GCMs compared to cold fronts.

## 5 Conclusions

In this study, we evaluated how climate models from the CMIP6 and EURO-CORDEX ensembles represent atmospheric frontal
frequencies, cross-frontal structures, and associated extreme precipitation, using ERA5 as a reference. To ensure consistency,
all model results were compared with ERA5 remapped to grids with comparable spatial resolution, thus reducing resolution-
induced detection and precipitation biases.



All models exhibit substantial biases in the cold and warm front frequencies, with negative biases over the North Atlantic and positive biases over continental regions. These patterns may result from a combination of an overly zonal storm track and enhanced frontogenesis arising from reduced boundary-layer friction. Enhanced frontogenesis would strengthen $\nabla\theta_e$, which could raise the probability of front detection by our identification scheme.

The composite analysis of frontal cross-sections reveals that the synoptic fields and circulation are well captured. The frontal structure is generally well represented, but smaller-scale features, such as gradients and mesoscale circulation show a strong dependence on resolution. Although this effect may seem obvious, it contributes to the underrepresentation of mesoscale dynamics that influence extreme precipitation, particularly in cold frontal systems. These systems are driven by smaller-scale processes, whereas warm fronts, due to their larger spatial extent, are governed by more synoptic-scale dynamics, thus benefiting less from higher resolutions.

The strong resolution sensitivity evident in the composite analysis is also reflected in the representation of frontal precipitation. Higher model resolution leads to increased cold frontal precipitation but has little effect on warm frontal precipitation. Although the more frequent detection of cold fronts, especially near occlusion points, may contribute to the apparent increase in cold frontal precipitation with resolution, this is likely not the only cause. In general, cold frontal extreme precipitation tends to be underestimated, whereas warm frontal precipitation is overestimated. The contrast between cold and warm frontal precipitation likely reflects the broader spatial extent and stronger synoptic-scale control of warm frontal systems, in contrast to the smaller-scale processes that govern cold fronts. The added value of higher resolution appears to arise from improved representation of these smaller-scale cold frontal processes.

Our findings highlight the importance of model resolution in simulating frontal dynamics and associated extremes. Identifying and understanding the sources of these biases will increase confidence in future climate projections. Since cold frontal extremes are more sensitive to resolution, projections from coarse-resolution models may underestimate future intensification, whereas warm frontal extremes are likely more robust to grid size.

*Code and data availability.* The ERA5 dataset is available at the Copernicus Climate Data Store. The CMIP6 and EURO-CORDEX data was downloaded from the Earth System Grid Federation (ESGF) nodes. The code used in this study is available on request from the corresponding author.



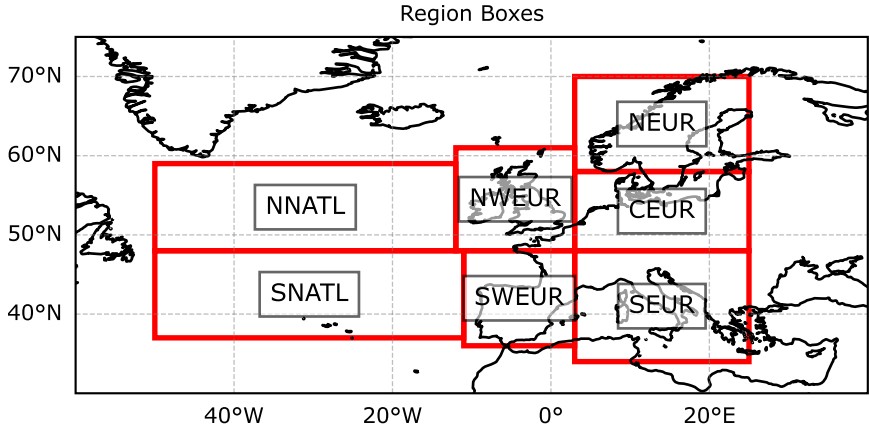

**Figure A1.** Regions used for regressing mean front frequency (Fig. 3) and mean frontal precipitation (Fig. 6) with resolution. Abbreviations stand for Northern North Atlantic (NNATL), Southern North Atlantic (SNATL), North-Western Europe (NWEUR), South-Western Europe (SWEUR), Northern Europe (NEUR), Central Europe (CEUR) and Southern Europe (SEUR), with the first two only being used when excluding CORDEX, because they are outside the domain.

**Appendix A**



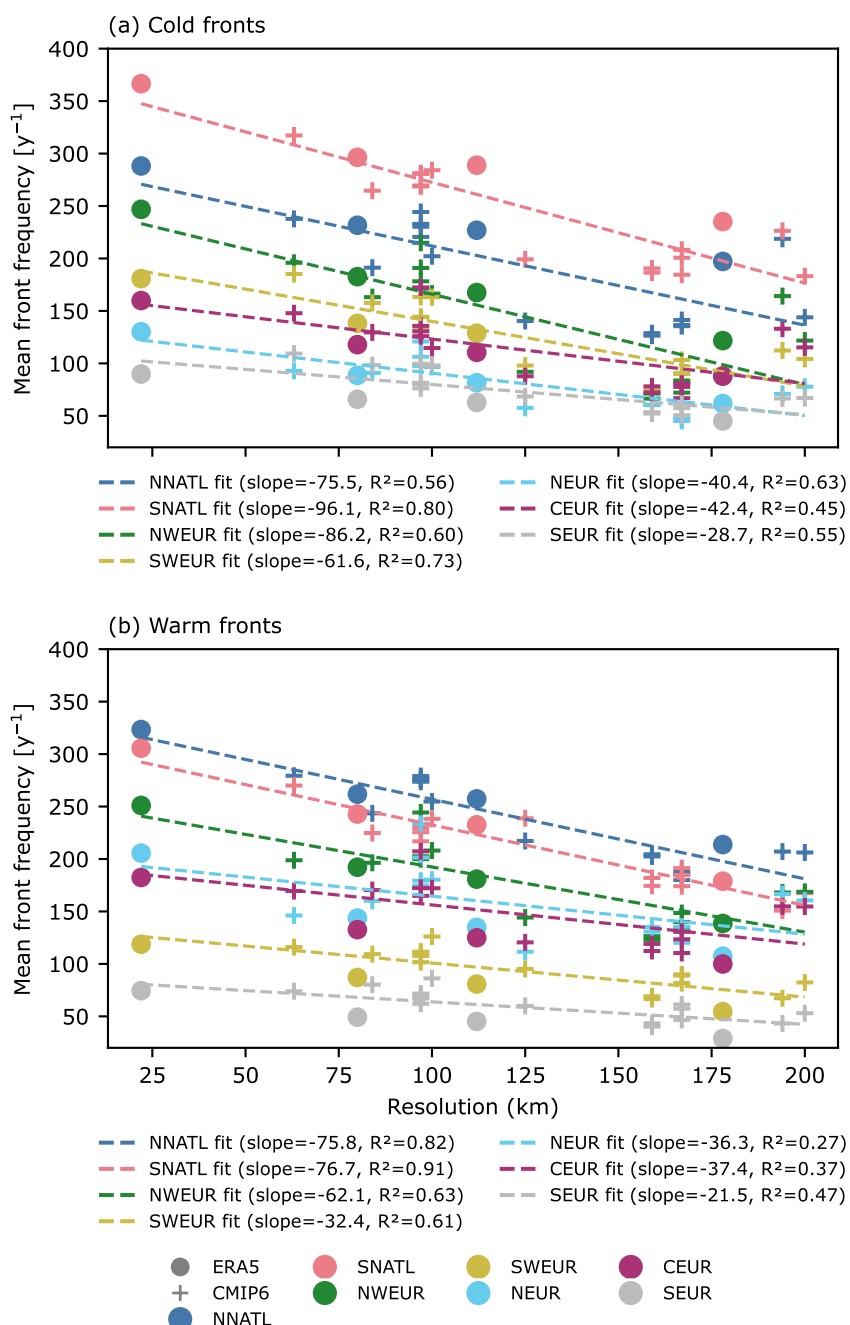

**Figure A2.** Same as Fig. 3, but without the EURO-CORDEX models.



**Figure A3.** Same as Fig. 4, but for total extreme precipitation.




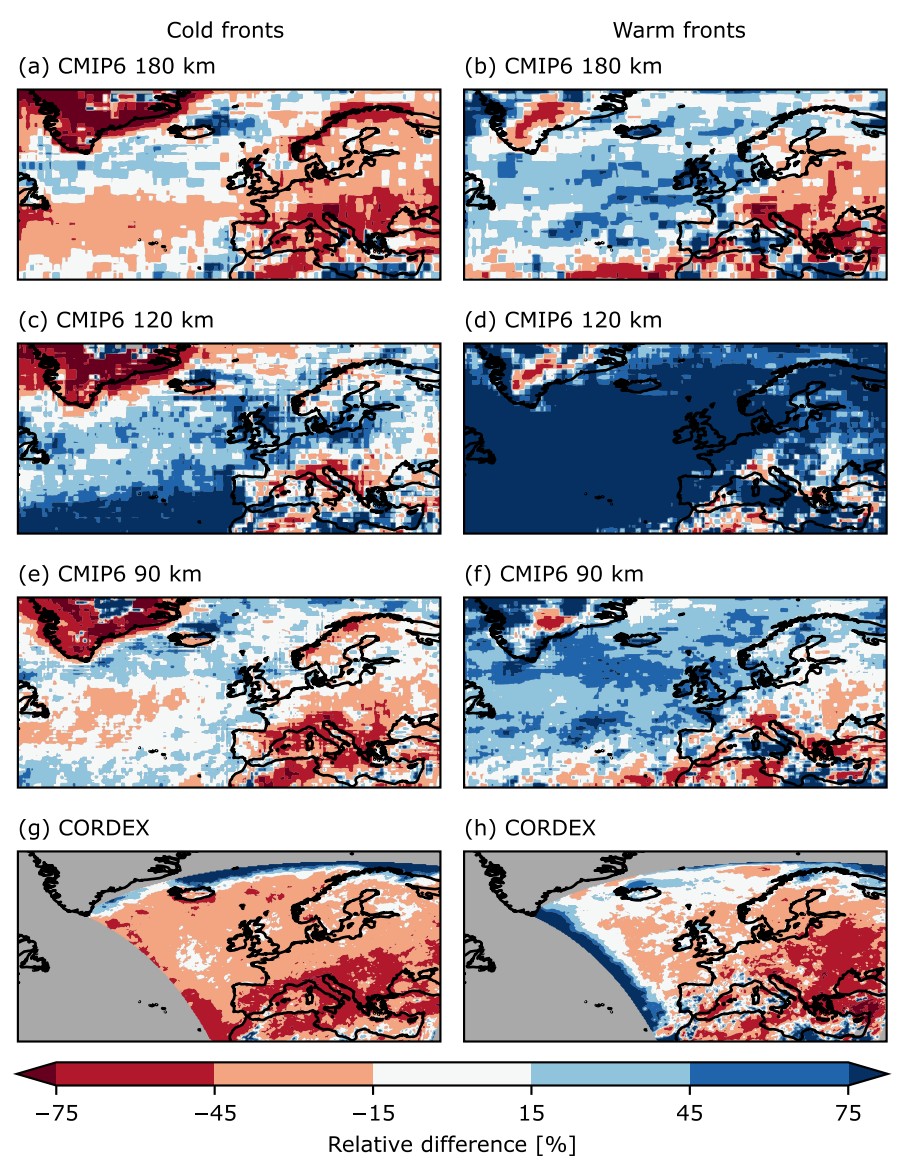

**Figure A4.** Same as Fig. 4, but for extreme precipitation per front.



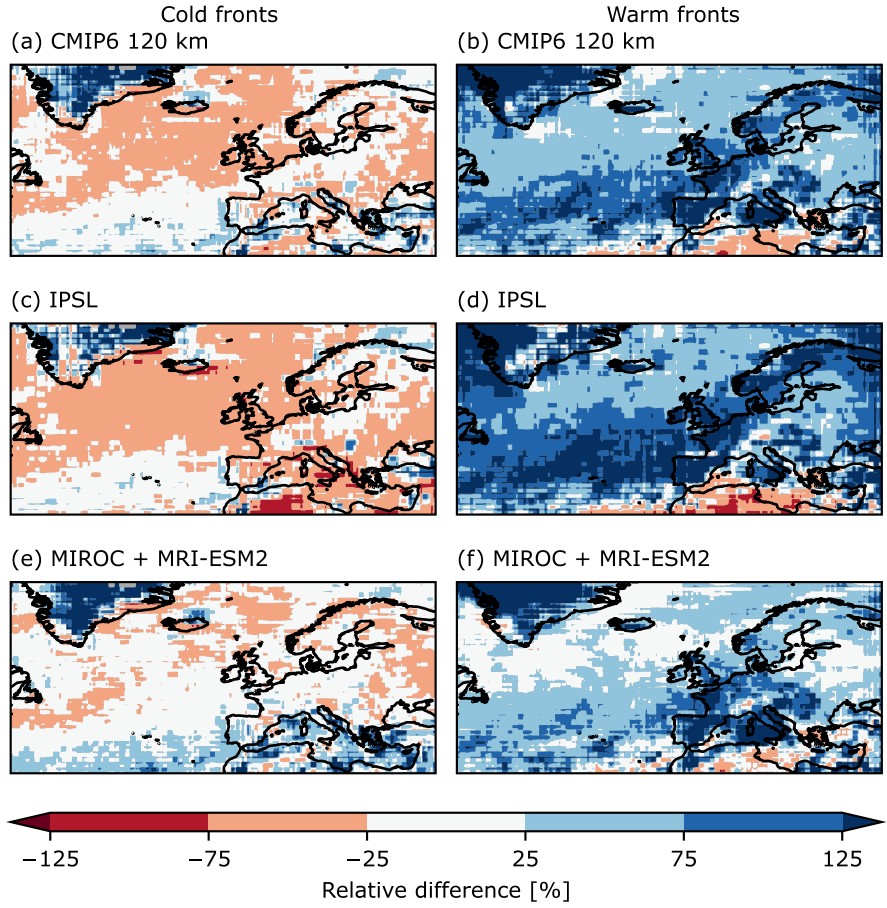

**Figure A5.** Same as Fig. 4, but for (a–b) CMIP6 120 km sub-ensemble and the contributing models of (c–d) IPSL and (e–f) MIROC and MRI-ESM2. The IPSL models have high latitudinal but low longitudinal resolution. This configuration may favor the representation of warm fronts, which tend to extend zonally and propagate more meridionally than cold fronts. On average the warm frontal cross-section may thus be better resolved, leading to a higher fraction of total extreme precipitation. This is supported by the fact that the IPSL models produce less cold frontal extreme precipitation and substantially more warm frontal precipitation than the rest of the sub-ensemble.



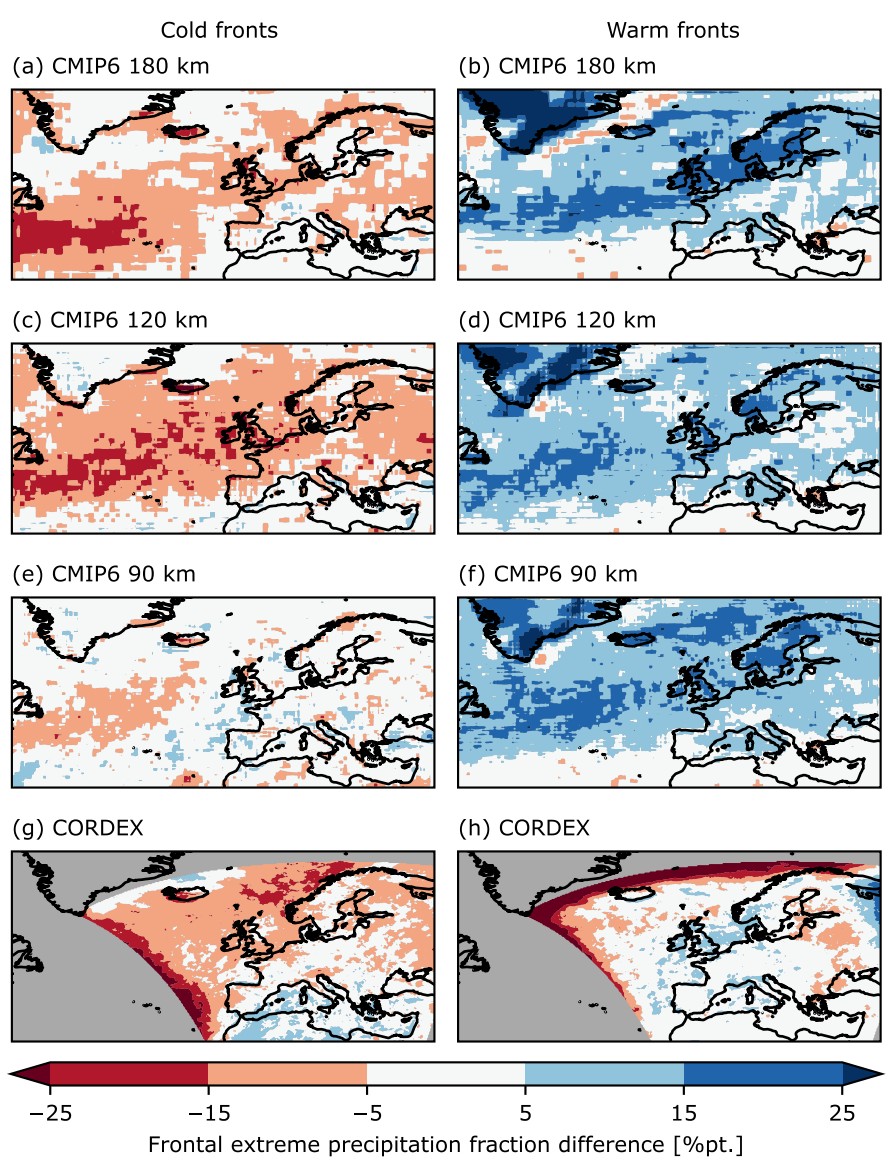

**Figure A6.** Same as Fig. 5, but percentage point difference of frontal extreme precipitation fraction.



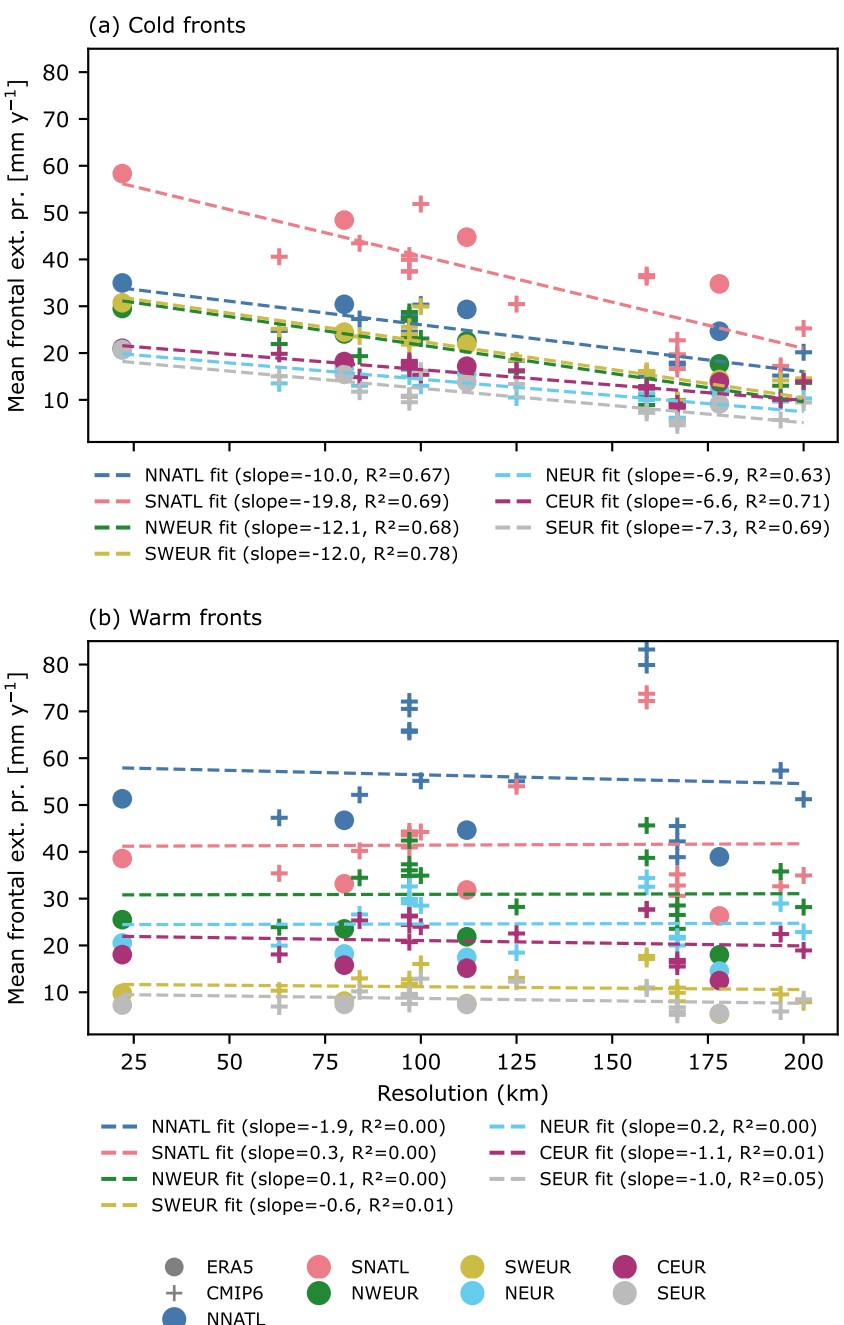

**Figure A7.** Same as Fig. 6, but without the EURO-CORDEX models. For warm fronts, the regression does not capture the behavior of the ERA5 data, which shows a weak negative relation. If the IPSL models (x = 159 km), which exhibit the highest values, are excluded, the regression aligns more with the ERA5 data, again highlighting the effect of higher latitudinal resolution on warm frontal extreme precipitation.



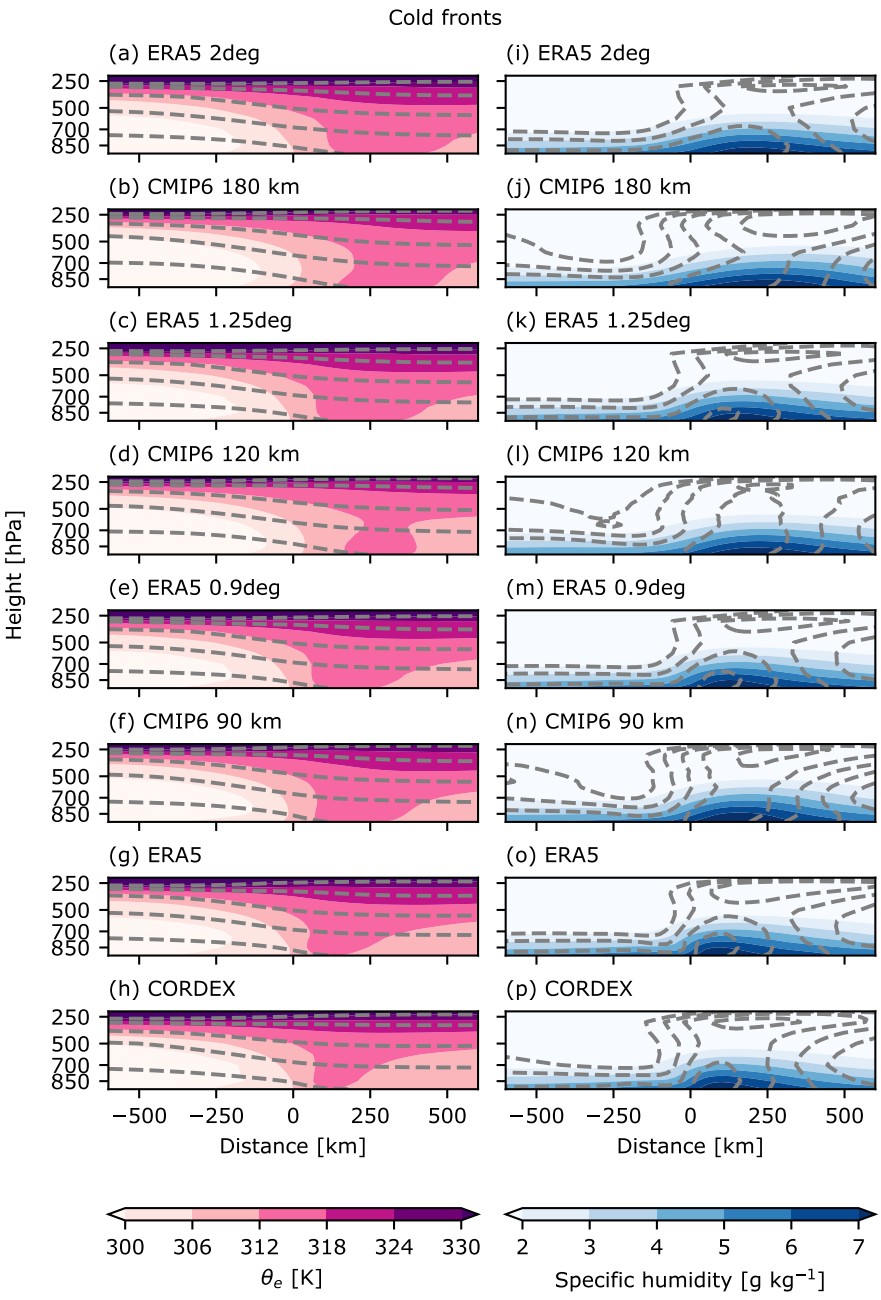

**Figure A8.** Same as Fig. 7, but colored contours display (a–h) $\theta_e$ and (i–p) specific humidity. Contours indicate (a–h) $\theta$ from $290-330$ K in 10 K steps and (i-p) relative humidity from 60–100 % in 10 % steps.





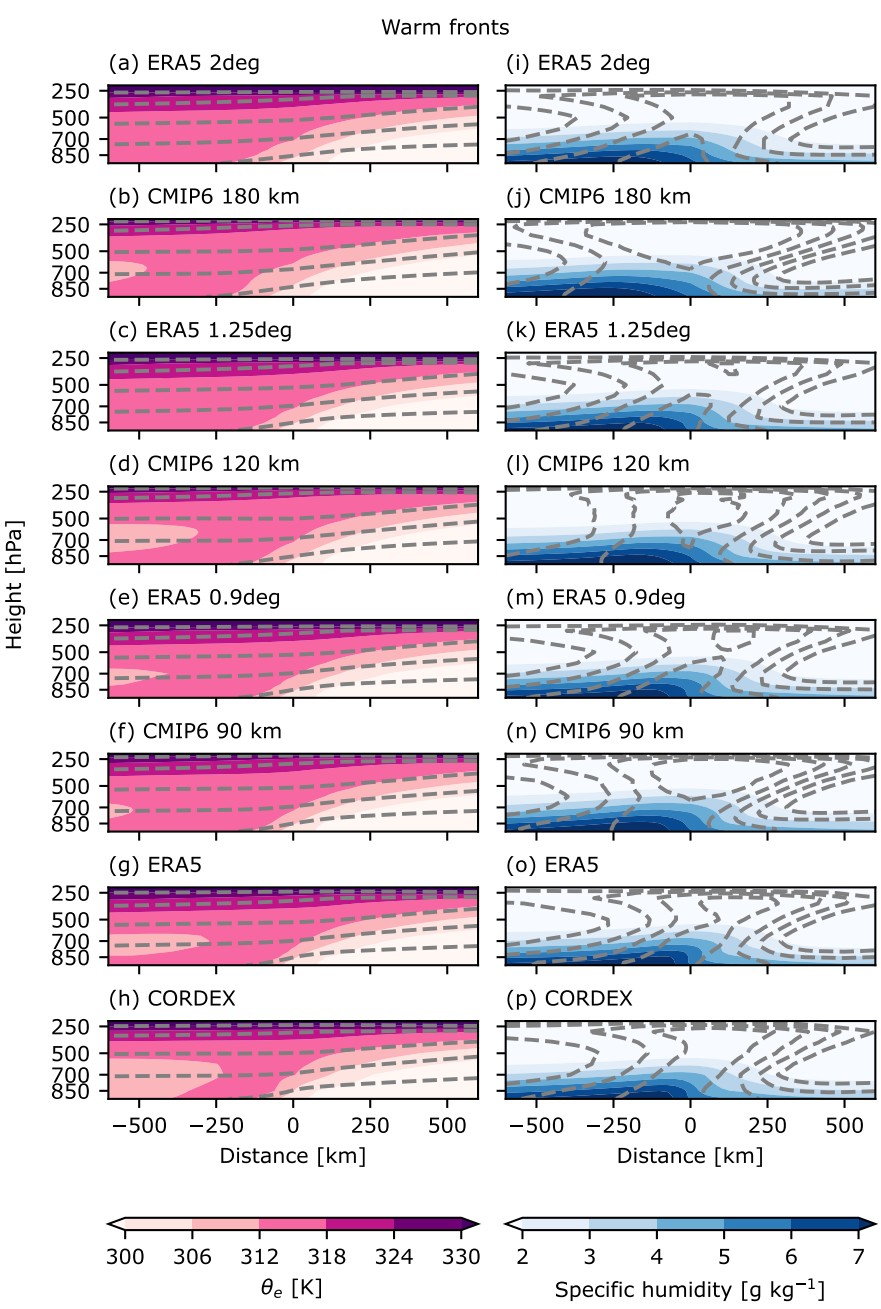

**Figure A9.** Same as Fig. A8, but for warm frontal cross-sections.





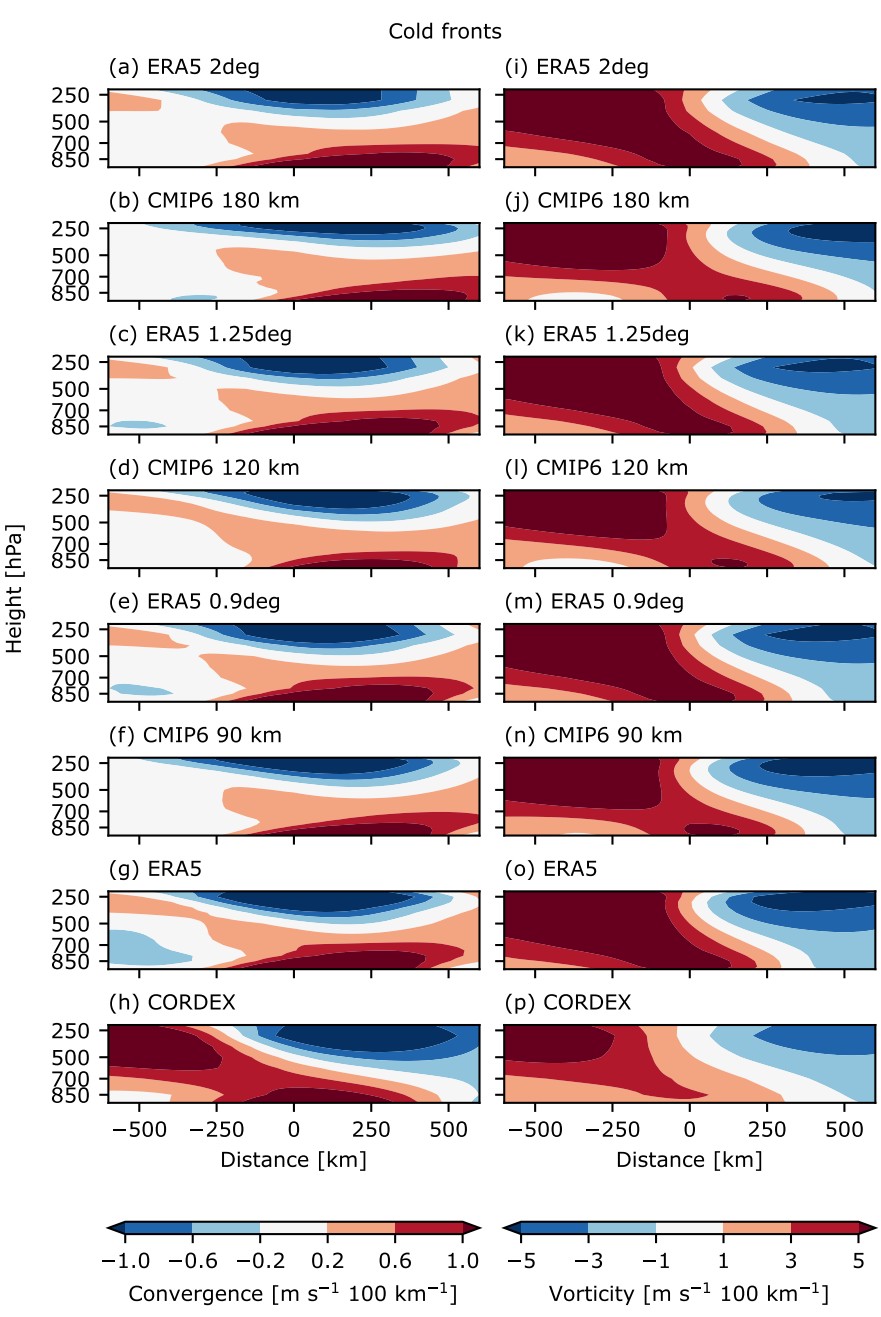

**Figure A10.** Same as Fig. 7, but colored contours display synoptic (a–h) convergence and (i–p) vorticity.




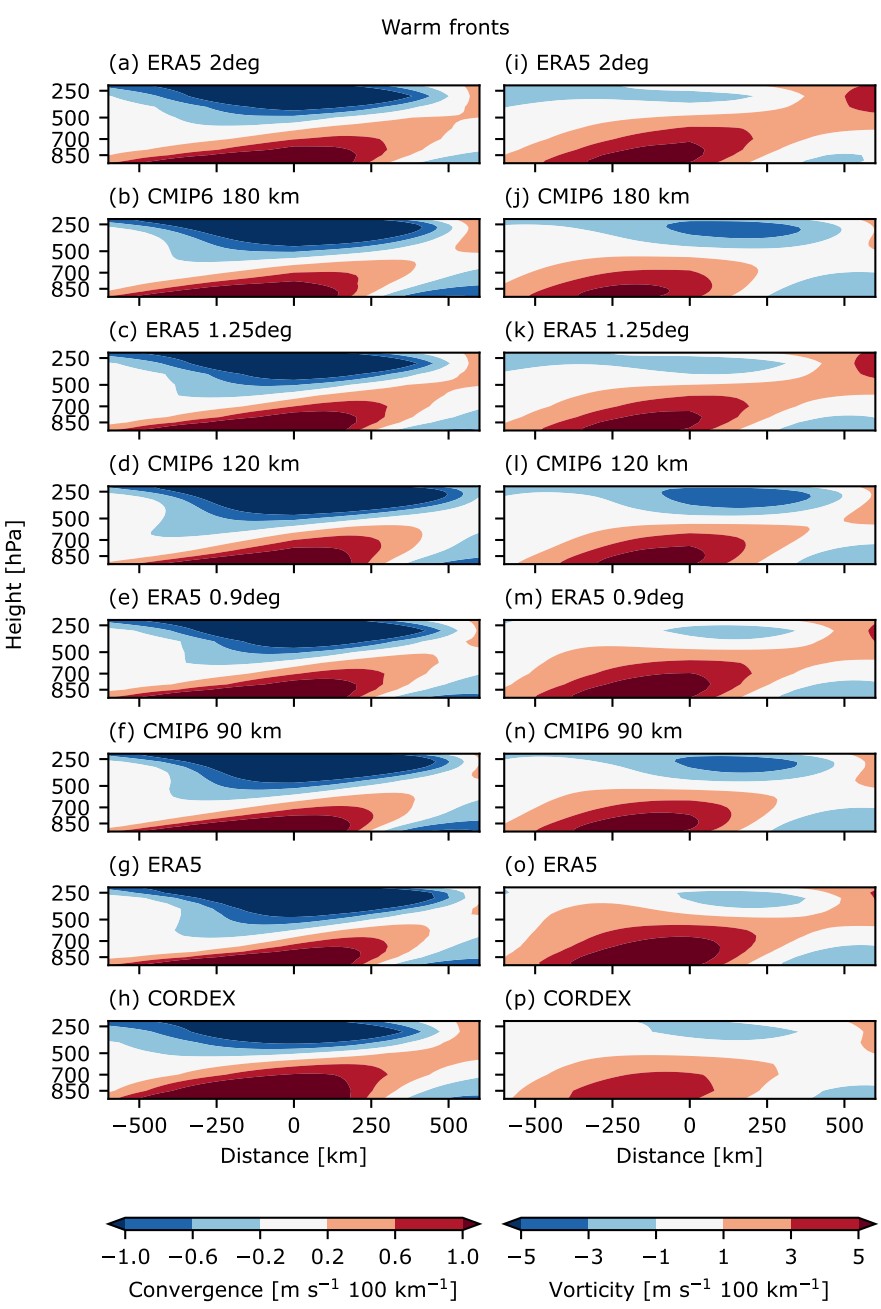

**Figure A11.** Same as Fig. A10, but for warm frontal cross-sections.



*Author contributions.*  AS developed the methodology, carried out the data acquisition and analysis, and wrote the manuscript. DM conceived the project idea. All authors contributed to the interpretation of the results and provided feedback on the manuscript.

*Competing interests.*  The authors declare that they have no conflict of interest.

*Acknowledgements.*  This research was funded by the Austrian Science Fund (FWF) in course of the INTERACT Project (Interactions across
scales shaping frontal weather extremes in a changing climate) (I 4831-N). We further want to thank our scientific advisory board member
Stephan Pfahl for supporting our research.



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
