# Peer review of "Resolution Dependence and Biases in Cold and Warm Frontal Extreme Precipitation over Europe in CMIP6 and EURO-CORDEX Models"

_EGUsphere, 2025_

## Editor Comment (EC1)

**Resolution dependence and biases in cold and warm frontal extreme precipitation over Europe in CMIP6 and EURO-CORDEX models**

by Armin Schaffer, Tobias Lichtenegger, Albert Ossó, and Douglas Maraun

**Dear Armin Schaffer**

Thank you for already addressing the reviewers' comments in your final author comments. Both reviewers are very positive about the merit of your work and addressing their comments will make the paper even clearer and stronger. I have few additional points, which I invite you to consider when preparing the revised version of your paper:

- General remark: you currently place many figures in the Appendix, and some paragraphs discuss mainly figures that the reader finds in the Appendix. I wonder whether this is ideal for the reader, because they have to flip many pages back and forth when reading the text and looking at the figures. I wonder whether it would not be better to put these additional figures into a Supplement. It is easier, in my view, for the reader to have two documents open, the main paper and the supplement than to navigate between text and appendix. Maybe Fig. A1 is an exception, this figure in fine in the appendix.
- L9: not sure that "underrepresentation" is the best term here, I would write "misrepresentation" or "inadequate representation".
- L13: maybe better "larger horizontal scales" are WF are typically shallower than CF.
- L30: please put references in chronological order.
- L33: I would use here past tense "we examined ... and analyzed ...".
- L45: maybe better "... in a study based on climate model simulations."
- P3 footnote 4, I don't understand "with geopotential missing 725-275 (50 hPa steps)". Do you have no geopotential date between 725 and 275 hPa? What is meant then by 50 hPa steps?
- L61: correct formatting of references (years should not be in (...)).
- Section 3.1: I appreciate that you try to be brief here and not repeat too much information from your previous paper. However, currently it is unclear what the TFP is → either explain or skip entirely. Also, I recommend that you reference the origin of this front detection method (I assume it goes back to some studies by Tim Hewson).
- L92: not sure that the term "back-bent occlusion fronts" is standard. First, I would write only "occlusions" instead of "occlusion fronts", and, more importantly, I think the original idea by Shapiro and Keyser when introducing the concept of the bent-back (warm) front was that this type of front is not an occlusion. I would write "exclude bent-back fronts from ...".
- L93: not clear to me what you regard as the warm conveyor belt here.

- L106: consistent with a reviewer comments, I would not call a return period of 50 days as "extreme". Maybe "intense/heavy precipitation events" would be more appropriate.
- L107: this reads complicated, can you explain this better such that others could reproduce your approach?
- L113: not clear what "standardized" means here.
- L116: please remind the reader what these regions are it is not convenient if the reader has to search for important details in another paper.
- Figures 7 and 8: this is, in my view, the highlight and main novelty of your study. It is therefore a pity that the panels are so small. You currently use a lot of space for the headings (e.g., "ERA5 2deg". I suggest that you place the labels next to the panels (instead of on top) and thereby you can enlarge the vertical dimension of your panels.
- L224: please give some indication how these extreme fronts have been selected.
- L230: my view on these interesting results is that the differences between ERA5 and the climate models are substantial, in particular for vorticity in cold fronts (Fig. 7). Please discuss more whether this could be an effect of vertical resolution. How do the vorticity cross sections look like for ERA5 if you only use the limited vertical levels available for some of the climate models? Why is vorticity particularly "wrong" near 500-700 hPa?
- L254: I find this concluding statement a bit too positive, given the results shown in Fig. 7.
- I don't think that Figs. A8-A11 are discussed in the text. In particular, I have a hard time understanding Figs. A10 and A11 in comparison with Figs. 7 and 8: what are these "synoptic" vorticity and convergence fields? I suggest removing them from the paper, as they were not discussed in the text and therefore not considered by the reviewers.

Looking forward to receiving the revised version of your paper. With best regards, Heini Wernli

---

## Author Response (AR1)

**Resolution Dependence and Biases in Cold and Warm Frontal Extreme Precipitation over Europe in CMIP6 and EURO-CORDEX Models**

**Response to Referee #1**

We thank the referee for the helpful feedback and suggestions. Below, we address each comment in detail.

1. Table 1: Could you use symbols such as *, # etc for the footnotes? It looks a bit confusing having the superscript numbers.

We adapted the footnote style to lower case superscript letters, following the journal guidelines.

2. Line 63: Could you make it clear here whether you do the regridding before the front identification?

We edited the sentence to make it clear that the regridding was done before any analysis was performed: *"…by following a similar approach to that described in Volosciuk et al. (2015): **prior to the front detection and analysis,** ERA5 is remapped to the three aforementioned coarser resolutions of 2° × 2°, 1.25° × 1.25°, and 0.9° × 0.9°."*

3. Line 101: I'm not sure what is meant by "Using on a grid-factor".

We removed the *"on"* in the sentence: *"Using a grid-factor (e.g. number of high-resolution grid points per low-resolution grid point) …"*. We want to convey that we cannot use a factor (e.g. 0.25° grid vs 1° gird = factor 16) to normalize the front frequency, because within the finer grid the number of frontal pixels can vary (due to frontal extend, curvature, etc.) and thus cannot be accounted for with a constant factor.

4. Line 103-104: I'm not sure how the resulting objects can have consistent width throughout the data. Although the radius is 300km, it depends on there being grid points within that radius in order to provide information. Could you clarify this?

That is correct. The 300 km radius used to define the frontal area masks all grid points whose centers fall within the circle. Consequently, the number of grid points within each frontal area depends on the model resolution. We acknowledge that this approach would become less reliable for very coarse data. However, our lowest-resolution model has a horizontal grid spacing of roughly 200 km, meaning that the frontal area spans on

average about three grid points (≈ 600 km in width), which we consider sufficient for our analysis.

Since the radius is defined in physical space, the resulting frontal objects represent the same spatial extent across models, making their frequencies as comparable as possible. To clarify this for the reader, we have revised the sentence to: *"The resulting objects have **a consistent physical width of 600 km across all datasets, independent of their resolution**."*

5. Line 154: This seems inconsistent with the next paragraph that the cold fronts are more sensitive to resolution than the warm fronts.

We agree that the statement seems contradictory. The statement refers to the overall bias favoring warm fronts, which remains consistent across model resolutions. The effect described in the following paragraph—that cold front detection is more sensitive to resolution—is relatively small compared to this bias. We changed the statement to: *"Comparing cold and warm fronts in more detail reveals that warm fronts are generally detected more frequently in the models. **This overall bias is consistent across all model resolutions.**"*

6. Line 162: Is this bias in both cold and warm fronts? Can you reference the figure panels here?

The bias is present for both cold and warm fronts alike. Previously we only mentioned the regression slopes for cold fronts in the norther regions (NEUR, NWEUR). We now added the warm front values in parentheses and added the references to the corresponding Figures: *"When CORDEX is excluded, the relationship becomes stronger, with the steepness of the slope increasing from 54 and 16 **(49 and 17) (Fig. 3)** to 86 and 40 **(76 and 36) (Fig. A2)** additional fronts per 100 km **for cold (warm) fronts, for NWEUR and NEUR** respectively."*

7. Figure 5: How much of the bias here is due to the precipitation values themselves, rather than the allocation to the fronts? Since you are looking at the total extreme precipitation values this could be an issue. Often analysis focuses on the proportion of extreme precipitation associated with fronts, which somewhat reduces this impact.

We agree that looking into precipitation values alone can be misleading. A paragraph on frontal extreme precipitation fractions, along with a corresponding figure in the Appendix (Fig. A6), is already included. In Fig. A6, the percentage point difference relative to ERA5 decreases for cold fronts but remains largely constant for warm fronts in CMIP6,

supporting our conclusion that cold fronts benefit more from higher resolutions. We further acknowledge the impact of resolution on the total extreme precipitation in ERA5 in Fig. A3.

8.  Line 195: Are you still looking at figure 5 in this paragraph?

Yes. We added references to Fig. 5 to make this more apparent.

9.  Line 199: does this mean higher than ERA5? Can you be clear in the text what you are comparing between?

For clarity, we have added that the warm frontal precipitation over the Atlantic is compared to ERA5.

10. Figure 7 a-p: In what way is this the "mesoscale" convergence and vorticity?

In our previous paper (Schaffer et al., 2024) we explain in detail how we split the synoptic and mesoscale dynamics. We added following sentences to the composite methodology: "**To *analyze front relative circulation composites, the dynamic variable fields are split into the synoptic- and mesoscale using a spectral filter. Wavelengths longer than 1000 km make up the synoptic- and shorter wavelengths the mesoscale.*"** In Fig. 7a-p we display the horizontal convergence and vorticity of only the short wavelength wind field. We further added the approximate location of the frontal surfaces (based on the TFP zero-contour of the composites). We think this further increases the intuitive understanding of the cross-section figures.

11. Figure 7: I don't think it is discussed what the impact of the vertical resolution of the available data is on these patterns.

In theory, the vertical resolution could influence the representation of circulation patterns. Before plotting, all datasets are bilinearly interpolated to the same standard pressure levels. Since the resulting composites are already quite smooth, this interpolation has no significant impact. Even in the most extreme case—the ALADIN63 simulations, which provide only five vertical levels (925, 850, 700, 500, and 200 hPa)—the key atmospheric layers relevant for representing the frontal structure are still adequately captured. We added a sentence I the methodology section: *"The extracted fields are rotated into the cross-frontal direction **and bilinearly interpolated to a common set of standard pressure levels (925-200 hPa in 25 hPa steps) before computing the composites. While differences in the native vertical resolution could affect the representation of the frontal structure, the resulting composites are**

*smooth, suggesting that interpolation and vertical resolution differences have only a minor impact."*

12. Line 230: What figure is being referred to here?

We added references to Fig. 7 and 8: *"Overall, the **cold (Fig. 7**) and **warm frontal structure (Fig. 8)** is well captured across all models ..."*

13. Line 231: Is this compared to ERA5?

The biases are always evaluated by comparing the sub-ensemble mean to ERA5 with similar grid spacing.

14. Line 232: Where is the data coming from for this -30% to 30% statement?

This value range is derived by computing the relative difference of maximum convergence of each model sub-ensemble and ERA5 with similar grid spacing. We present these percentages to give a rough estimate of how well not only the structure, but also the peak values of the circulation are represented by the models.

We hope that our revisions have improved the clarity of the manuscript and helped readers to better understand our study.

Best regards,
Armin Schaffer

**Response to Referee #2**

We thank the referee for the helpful feedback and suggestions. Below, we address each comment in detail.

- Table 1: even with the footnote, I'm unclear what the two numbers of vertical levels for the CMIP6 models mean - I think that you're implying that geopotential height wasn't available for 10 of the levels the other fields were and thus only has 50 hPa spacing between 725 and 275 hPa.

This is correct. All variables except geopotential are available at 33 levels. Geopotential is missing 10 levels in the 725-275 hPa range. For clarity, we have revised the footnote to: *"1000-200 (25 hPa steps), with geopotential **missing 10 levels** (725-275 in 50 hPa steps)"*

- Line 48: Is there some reason for this particular selection of models from the CMIP6 ensemble, particularly with the outsized influence of IPSL-CM6A-LR and IPSL-CM6A-LR-INCA on the mid-resolution sub-ensemble?

First, we analyzed all CMIP6 models for which the necessary data were available. The models were grouped into sub-ensembles based on their horizontal resolutions, which roughly cluster into three groups. The IPSL models are outliers due to their substantially different longitudinal and latitudinal grid spacing. We included these models in the mid-resolution subset because their front frequency and precipitation patterns show higher correlation with this subset than with the low-resolution sub-ensemble. This behavior is likely related to the fact that fronts (particularly warm fronts) are more sensitive to latitudinal than longitudinal resolution, as discussed in Fig. A5.

- Line 56-57: please briefly describe the nature of the spectral filter - what is it filtering out?

We agree that the smoothing effect of the spectral filter is not clearly pointed out. We changed the sentence to: *"To improve consistency, all data are spectrally filtered prior to the analysis **to reduce small-scale variability.**"* For more details we refer the reader to our previous study (Schaffer et al., 2024), where we discuss the spectral filter in detail.

- Line 69: is the remapping to 0.25 degree resolution for plotting/comparison a conservative remapping and not a bilinear one? A quick specification would be helpful.

We added: *"... all results are remapped **conservatively** ..."*, to clarify the remapping method applied.

- Line 89: whether or not these are "false" cold fronts is a matter of definition - I understand they aren't relevant for an examination of fronts associated with mobile systems instead of those coming from persistent land-sea contrasts though!

We fully agree with this statement. What constitutes a front depends strongly on its definition. Accordingly, we have revised the phrasing from *"...the detection of false cold fronts..."* to *"...the detection of **spurious air mass boundaries as cold fronts**..."*

- Line 106: Is 6-hourly rainfall above the 99.5th percentile suitably strong enough to be considered extreme, especially when that corresponds to an expected return period of ~ 50 days?

We carefully considered the choice of threshold. What is considered "truly extreme" depends strongly on the chosen definition and the specific focus of the analysis. In the present study, we adopted the 99th percentile definition of extreme precipitation from Catto & Pfahl (2014) as a baseline and increased the threshold to represent events that are twice as rare. In our next study, we plan to classify precipitation above the 99.5th percentile as heavy and above the 99.95th percentile as extreme, corresponding to 50-day and 500-day return periods, respectively. Whether a 500-day return period can be considered "sufficiently extreme" remains subject to interpretation.

- Line 112-114: a figure might help to illustrate what is considered a frontal object for the compositing by this definition.

There is no straightforward way to illustrate the composite method in a figure. To improve clarity, we have rewritten the description of the composite method in the manuscript, presenting it in a more intuitive, stepwise manner for the reader:
*"**To construct the frontal composites, we apply a multi-step procedure designed to capture the typical structures of intense frontal systems. First, we determine the precipitation associated with each frontal object by focusing on the most active 200 km segment of the front. To do so, we calculate the number of frontal grid points corresponding to a length of approximately 200 km based on the model resolution. All frontal points within a given object are then ranked by their standardized**"*

*precipitation values, and the top-ranked points that together represent about 200 km of frontal length are selected. The precipitation of the frontal object is defined as the mean precipitation of these selected frontal points. This approach ensures that the composites are based on the most intense and meteorologically relevant parts of each front, independent of variations in model resolution or front length."*

- Section 4.2: is the "frontal extreme precipitation" extreme precipitation (as in section 3.2) associated with fronts, or the precipitation associated with extreme fronts (as in section 3.3)?

Frontal extreme precipitation is defined as described in Section 3.2. At the beginning of Section 4.3, we note this explicitly for the composite analysis: *"Note that these composites are based on fronts selected independently from those associated with the extreme precipitation events discussed in the previous section..."* To make this clearer in Section 4.2, we have added a reference to the definition in the methodology.

We hope that our revisions have improved the clarity of the manuscript and helped readers to better understand our study.

Best regards,
Armin Schaffer

**Response to the Editor**

Dear Heini Wernli,

We again thank you for your constructive feedback. Below we address each of your comments in detail and highlight the changes to the manuscript.

• General remark: you currently place many figures in the Appendix, and some paragraphs discuss mainly figures that the reader finds in the Appendix. I wonder whether this is ideal for the reader, because they have to flip many pages back and forth when reading the text and looking at the figures. I wonder whether it would not be better to put these additional figures into a Supplement. It is easier, in my view, for the reader to have two documents open, the main paper and the supplement than to navigate between text and appendix. Maybe Fig. A1 is an exception, this figure in fine in the appendix.

We moved Figure A2-A11 to a separate Supplementary Information file (Figs. S1-S10).

• L9: not sure that "underrepresentation" is the best term here, I would write "misrepresentation" or "inadequate representation".

We agree and change the wording to "inadequate representation".

• L13: maybe better "larger horizontal scales" are WF are typically shallower than CF.

We agree, "larger horizonal scales" is more accurate description of the fact that warm fronts are shallower and thus extend further.

• L30: please put references in chronological order.

We changed this and one more citation to be in chronological order.

• L33: I would use here past tense "we examined ... and analyzed ...".

We adapted the sentence accordingly.

• L45: maybe better "... in a study based on climate model simulations."

We agree. This makes the sentence more readable.

• P3 footnote 4, I don't understand "with geopotential missing 725-275 (50 hPa steps)". Do you have no geopotential date between 725 and 275 hPa? What is meant then by 50 hPa steps?

We already addressed this point in a reviewer comment. The levels 725, 675, 625, … 325, 275 hPa (10 levels) do not have geopotential data. We used CMIP6 model levels to generate highly resolved pressure level output, but geopotential is not available on model levels on the ESGF servers. To make this even more clear, we changed the footnote to: "1000-200 (25 hPa steps), **with 10 levels missing geopotential data (725, 675, 625, 575, 525, 475, 425, 375, 325, 275 hPa)**"

• L61: correct formatting of references (years should not be in (…)).

We corrected the citation style.

• Section 3.1: I appreciate that you try to be brief here and not repeat too much information from your previous paper. However, currently it is unclear what the TFP is à either explain or skip entirely. Also, I recommend that you reference the origin of this front detection method (I assume it goes back to some studies by Tim Hewson).

We included are more detailed description of TFP: "The final frontal points are identified as local maxima in $\nabla\theta e$ where the Thermal Front Parameter (TFP) is closest to zero. **TFP is defined as: TFP = $- \nabla|\nabla\theta e| \cdot \nabla\theta e|\nabla\theta e|$ and measures the rate of change of $\nabla\theta e$ in direction of the gradient.**"
Our front detection method combines multiple methods from different studies with the most important ones being Hewson (1998) & Jenkner et al. (2009) and was developed by multiple people in our institute (e.g. Ritter, 2014). We discussed the choice of method in detail in our previous paper. This is why we did not include this discussion in the current manuscript. However, we now acknowledge the groundwork these studies performed by starting the section with: "**The front detection scheme employed in this study follows established approaches in the literature (e.g., Hewson, 1998; Jenkner et al., 2009), in which regions of strong thermal gradients are identified and a wind–based threshold is subsequently applied.**"

• L92: not sure that the term "back-bent occlusion fronts" is standard. First, I would write only "occlusions" instead of "occlusion fronts", and, more importantly, I think the original idea by Shapiro and Keyser when introducing the concept of the bent-back (warm) front was that this type of front is not an occlusion. I would write "exclude bent-back fronts from …".

As far as we know the terms "back-bent occlusion", "back-bent front" and "back-bent warm front" are describing the same phenomenon. "back-bent front" seems to be the most widely used. We changed the text accordingly.

• L93: not clear to me what you regard as the warm conveyor belt here.

In this context, we refer to the warm conveyor belt in the sense of Browning (1973, 1985), i.e., the warm and moist airstream ahead of the cold front in extratropical cyclones. When applying our front detection, we occasionally detect humidity gradients on both flanks of this airstream. The gradients on the warm-side boundary are typically not associated with precipitation and are therefore not considered synoptic fronts for the purposes of our study. To address this, we apply a post-processing filter to remove these cases as much as possible.

• L106: consistent with a reviewer comments, I would not call a return period of 50 days as "extreme". Maybe "intense/heavy precipitation events" would be more appropriate.

As noted in our response to the reviewer, our initial definition followed Catto & Pfahl (2013), where "extreme" refers to 6-hourly precipitation above the 99th percentile. However, we agree that "heavy precipitation" is a more appropriate description for 50-day return period events in our context. We have therefore replaced "extreme precipitation" with "heavy precipitation" throughout the manuscript, including in the title: "Resolution Dependence and Biases in Cold and Warm Frontal **Heavy** Precipitation over Europe in CMIP6 and EURO-CORDEX Models".

• L107: this reads complicated, can you explain this better such that others could reproduce your approach?

We rewrote this paragraph and added an example to make it easier to understand: "Following Henin et al. (2019) precipitation is **split between cold and warm frontal when it falls within the 300 km radius** of both front types. In such cases, precipitation is partitioned **proportionally** to the number of grid points associated with each front **type (e.g., if 6 cold and 4 warm frontal points are within the radius, 60 % of the precipitation amount is classified as cold and 40 % as warm frontal)**."

• L113: not clear what "standardized" means here.

We added "...standardized precipitation values **(normalized by its mean and standard deviation)**, ...", to make our method clearer. We used standardized precipitation to minimize the effect of e.g. orography or coastlines, as described in our previous study.

• L116: please remind the reader what these regions are – it is not convenient if the reader has to search for important details in another paper.

We added the exact definition of the regions: "The regions used to evaluate these fronts **are defined as: Northwestern Europe (NWEUR, 48°N-61°N, 12°E-3°W), Southwestern Europe (SWEUR, 36°N-48°N, 11°E-4°W), and Central Europe (CEUR, 48°N-58°N, 3°W-25°W). These regions are selected due to their high front frequency and fraction of heavy frontal precipitation."**

• Figures 7 and 8: this is, in my view, the highlight and main novelty of your study. It is therefore a pity that the panels are so small. You currently use a lot of space for the headings (e.g., "ERA5 2deg". I suggest that you place the labels next to the panels (instead of on top) and thereby you can enlarge the vertical dimension of your panels.

We fully agree and changed the layout of all cross-section Figures accordingly. The panels still could be larger, but to do that we would have to split the two Figures into four, which we belief would clutter the manuscript.

• L224: please give some indication how these extreme fronts have been selected.

With this sentence we wanted to highlight that the composite and the heavy precipitation analysis are two independent analyses. The composite method is described in Section 3.3. We made this clearer by changing the citation to a reference of the section. We further changed the term "extreme fronts" to "strong fronts" to be more in line with the adapted definition of "heavy" precipitation: "For this purpose, we analyze cross-sections of composites **of strong** cold (Fig. 7) and warm fronts (Fig. 8). Note that these composites are based on fronts selected independently from those associated with the heavy precipitation discussed in the previous section, but by following the method described in **Section 3.3**."

• L230: my view on these interesting results is that the differences between ERA5 and the climate models are substantial, in particular for vorticity in cold fronts (Fig. 7). Please discuss more whether this could be an effect of vertical resolution. How do the vorticity cross sections look like for ERA5 if you only use the limited vertical levels available for some of the climate models? Why is vorticity particularly "wrong" near 500-700 hPa?

We do not believe that the mesoscale vorticity differences are primarily driven by vertical-resolution effects. CMIP6 data have higher vertical resolution than the ERA5 fields we use here, and one CORDEX model with relatively few vertical levels still

reproduces the sloping vorticity structure more closely to ERA5 than the coarser CMIP6 simulations.

Instead, the contrasting vorticity patterns are most likely linked to differences in the representation and position of the low-level jet. In the CMIP6 simulations, the low-level jet on average tends to be displaced farther ahead of the cold front (~450 km ahead of the cold front in CMIP6 vs ~300 km in ERA5), which shifts the upper- and lower-level vorticity maxima apart and produces a dipole-like vertical structure. In contrast, ERA5 composites show a smoother vertical transition because the low- and upper-level jets, and thus their associated vorticity maxima, are more closely aligned. The ERA5 data is the output of the high-resolution IFS model and therefore captures the low-level jet location more accurately than the CMIP6 models. Remapping ERA5 to coarser resolutions weakens the low-level jet but does not substantially change its mean position. We added the following to the text: "Mesoscale circulations, however, show some biases. **Vorticity in cold fronts (Fig. 7i – p) exhibits the biggest differences, with all CMIP6 sub-ensembles showing a split between the upper- and lower-level positive vorticity regions. In contrast, ERA5 has a continuous backwards-sloped area of high vorticity. This is due to the low-level jet position, which in CMIP6 on average is located further ahead of the cold front than in ERA5.**"

To further illustrate this, we here provide cold front composites of the cross- and along-frontal wind speed for ERA5 2° and CMIP6 180 km:

[Figure]

• L254: I find this concluding statement a bit too positive, given the results shown in Fig. 7.

This sentence refers to the large-scale fields (Fig. A8–A11 / Fig. S7–S10), not the mesoscale circulation. To avoid confusion, we clarified the wording as follows: "The composite analysis of frontal cross-sections reveals that the **large-scale fields and synoptic circulation** are well captured." With the additional figure references in the results section (see reply to the next comment), we believe the revised text now reads more intuitively.

• I don't think that Figs. A8-A11 are discussed in the text. In particular, I have a hard time understanding Figs. A10 and A11 in comparison with Figs. 7 and 8: what are these "synoptic" vorticity and convergence fields? I suggest removing them from the paper, as they were not discussed in the text and therefore not considered by the reviewers.

The content of the figures was already described in the text, but we had not included explicit references. We now added: "At the **large scale**, the composite fields of temperature, humidity **(Fig. S7–S8)**, and circulation **(Fig. S9–S10)** show good agreement between ERA5 and all model ensembles."
We believe these plots are valuable in the Supplementary Information, as they provide additional insight into the thermodynamic conditions and large-scale frontal circulation. As noted, there are no substantial differences between the reanalysis and the simulations, demonstrating that the large-scale environment of frontal systems is well represented even in coarse-resolution GCMs. Since the focus of our study is on the differences, we do not discuss the general frontal structure in detail here. A more detailed discussion of the cold frontal structure can be found in Schaffer et al. (2024).
In response to a reviewer suggestion, we also expanded the description of the synoptic and mesoscale decomposition in Section 3.3: **"To analyze composites of front-relative circulation, the dynamic variables are separated into synoptic and mesoscale components using a spectral filter, with wavelengths longer than 1000 km representing the synoptic scale and shorter wavelengths the mesoscale."**
Finally, we further added the approximate location of the frontal surfaces to all cross-section figures.

We believe that our revisions enhance the manuscript's clarity and provide readers with a clearer insight into our study.

Best regards,
Armin Schaffer